# FEW-SHOT LEARNING VIA LEARNING THE REPRESENTATION, PROVABLY

Simon S. Du[1*], Wei Hu[2*], Sham M. Kakade[1,3*], Jason D. Lee[2*], and Qi Lei[2*]

[1]University of Washington    [2]Princeton University    [3]Microsoft Research
{ssdu,sham}@cs.washington.edu, {huwei@cs.,
jasonlee@,qilei@}princeton.edu

## ABSTRACT

This paper studies few-shot learning via representation learning, where one uses $T$ source tasks with $n_1$ data per task to learn a representation in order to reduce the sample complexity of a target task for which there is only $n_2(\ll n_1)$ data. Specifically, we focus on the setting where there exists a good common representation between source and target, and our goal is to understand how much a sample size reduction is possible. First, we study the setting where this common representation is low-dimensional and provide a risk bound of $\tilde{O}(\frac{dk}{n_1 T} + \frac{k}{n_2})$ on the target task for the linear representation class; here $d$ is the ambient input dimension and $k(\ll d)$ is the dimension of the representation. This result bypasses the $\Omega(\frac{1}{T})$ barrier under the i.i.d. task assumption, and can capture the desired property that all $n_1 T$ samples from source tasks can be *pooled* together for representation learning. We further extend this result to handle a general representation function class and obtain a similar result. Next, we consider the setting where the common representation may be high-dimensional but is capacity-constrained (say in norm); here, we again demonstrate the advantage of representation learning in both high-dimensional linear regression and neural networks, and show that representation learning can fully utilize all $n_1 T$ samples from source tasks.

## 1 INTRODUCTION

A popular scheme for *few-shot learning*, i.e., learning in a data-scarce environment, is *representation learning*, where one first learns a feature extractor, or representation, e.g., the last layer of a convolutional neural network, from different but related source tasks, and then uses a simple predictor (usually a linear function) on top of this representation in the target task. The hope is that the learned representation captures the common structure across tasks, which makes a linear predictor sufficient for the target task. If the learned representation is good enough, it is possible that a few samples are sufficient for learning the target task, which can be much smaller than the number of samples required to learn the target task from scratch.

While representation learning has achieved tremendous success in a variety of applications (Bengio et al., 2013), its theoretical studies are limited. In existing theoretical work, the most natural algorithm is to explicitly look for the optimal representation given source data, which when combined with a (different) linear predictor on top for each task can achieve the smallest cumulative training error on the source tasks. Of course, it is not guaranteed that the representation found will be useful for the target task unless one makes some assumptions to characterize the connections between different tasks. Existing work often imposes a probabilistic assumption about the connection between tasks: each task is sampled i.i.d. from an underlying distribution. Under this assumption, Maurer et al. (2016) showed an $\tilde{O}(\frac{1}{\sqrt{T}} + \frac{1}{\sqrt{n_2}})$ risk bound on the target task, where $T$ is the number of source tasks, $n_1$ is the number of samples per source task, and $n_2$ is the number of samples from the target task.[1] Unsatisfactorily, this bound necessarily requires the number of tasks $T$ to be large, and it

---

[*]Alphabetical Order.

[1]We only focus on the dependence on $T$, $n_1$ and $n_2$ in this paragraph. Note that Maurer et al. (2016) only considered $n_1 = n_2$, but their approach does not give a better result even if $n_1 > n_2$.

does not improve when the number of samples per source task, $n_1$, increases. Intuitively, one should expect more data to help, and therefore an ideal bound would be $\frac{1}{\sqrt{n_1 T}} + \frac{1}{\sqrt{n_2}}$ (or $\frac{1}{n_1 T} + \frac{1}{n_2}$ in the realizable case), because $n_1 T$ is the total number of training data points from source tasks, which can be potentially pooled to learn the representation.

Unfortunately, as pointed out by Maurer et al. (2016), there exists an example that satisfies the i.i.d. task assumption for which $\Omega(\frac{1}{\sqrt{T}})$ is unavoidable (or $\Omega(\frac{1}{T})$ in the realizable setting). This means that *the i.i.d. assumption alone is not sufficient* if we want to take advantage of a large amount of samples per task. Therefore, a natural question is:

*What connections between tasks enable representation learning to utilize **all** source data?*

In this paper, we obtain the first set of results that fully utilize the $n_1 T$ data from source tasks. We replace the i.i.d. assumption over tasks with natural structural conditions on the input distributions and linear predictors. These conditions depict that the target task can be in some sense "covered" by the source tasks, which will further give rise to the desirable guarantees.

First, we study the setting where there exists a common well-specified *low-dimensional* representation in source and target tasks, and obtain an $\tilde{O}(\frac{dk}{n_1 T} + \frac{k}{n_2})$ risk bound on the target task where $d$ is the ambient input dimension, $k(\ll d)$ is the dimension of the representation, and $n_2$ is the number of data from the target task. Note that this improves the $\frac{d}{n_2}$ rate of just learning the target task without using representation learning. The term $\frac{dk}{n_1 T}$ indicates that we can fully exploit all $n_1 T$ data in the source tasks to learn the representation. We further extend this result to handle general representation function class and obtain an $\tilde{O}(\frac{\mathcal{C}(\Phi)}{n_1 T} + \frac{k}{n_2})$ risk bound on the target task, where $\Phi$ is the representation function class and $\mathcal{C}(\Phi)$ is a certain complexity measure of $\Phi$.

Second, we study the setting where there exists a common linear *high-dimensional* representation for source and target tasks, and obtain an $\tilde{O}\big(\frac{\bar{R}\sqrt{\mathrm{Tr}(\Sigma)}}{\sqrt{n_1 T}} + \frac{\bar{R}\sqrt{\|\Sigma\|_2}}{\sqrt{n_2}}\big)$ rate where $\bar{R}$ is a normalized nuclear norm control over linear predictors, and $\Sigma$ is the covariance matrix of the raw feature. This also improves over the baseline rate for the case without using representation learning. We further extend this result to two-layer neural networks with ReLU activation. Again, our results indicate that we can fully exploit $n_1 T$ source data.

A technical insight coming out of our analysis is that any capacity-controlled method that gets low test error on the source tasks must also get low test error on the target task by virtue of being forced to learn a good representation. Our result on high-dimensional representations shows that the capacity control for representation learning does not have to be through explicit low dimensionality.

**Organization.** The rest of the paper is organized as follows. We review related work in Section 2. In Section 3, we formally describe the setting we consider. In Section 4, we present our main result for low-dimensional linear representation learning. A generalization to nonlinear representation classes is demonstrated in Section 5. In Section 6, we present our main result for high-dimensional linear representation learning. In Section 7, we present our result for representation learning in neural networks. We conclude in Section 8 and leave most of the proofs to appendices.

## 2 RELATED WORK

The idea of multitask representation learning at least dates back to Caruana (1997); Thrun and Pratt (1998); Baxter (2000). Empirically, representation learning has shown its great power in various domains; see Bengio et al. (2013) for a survey. In particular, representation learning is widely adopted for few-shot learning tasks (Sun et al., 2017; Goyal et al., 2019). Representation learning is also closely connected to meta-learning (Schaul and Schmidhuber, 2010). Recent work Raghu et al. (2019) empirically suggested that the effectiveness of the popular meta-learning algorithm Model Agnostic Meta-Learning (MAML) is due to its ability to learn a useful representation. The scheme we analyze in this paper is closely related to Lee et al. (2019); Bertinetto et al. (2018) for meta-learning.

On the theoretical side, Baxter (2000) performed the first theoretical analysis and gave sample complexity bounds using covering numbers. Maurer et al. (2016) and follow-up work gave analyses on the benefit of representation learning for reducing the sample complexity of the target task. They assumed every task is i.i.d. drawn from an underlying distribution and can obtain an $\tilde{O}(\frac{1}{\sqrt{T}} + \frac{1}{\sqrt{n_2}})$

rate. As pointed out in Maurer et al. (2016), the $\frac{1}{\sqrt{T}}$ dependence is not improvable even if $n_1 \to \infty$ because $\frac{1}{\sqrt{T}}$ is the rate of concentration for the distribution over tasks.

The concurrent work of Tripuraneni et al. (2020) studies low-dimensional linear representation learning and obtains a similar result as ours in this case, but they assume isotropic inputs for all tasks, which is a special case of our result. Furthermore, we also provide results for high-dimensional linear representations, general non-linear representations, and two-layer neural networks. Tripuraneni et al. (2020) also give a computationally efficient algorithm for standard Gaussian inputs and a lower bound for subspace recovery in the low-dimensional linear setting.

Another recent line of theoretical work analyzed gradient-based meta-learning methods (Denevi et al., 2019; Finn et al., 2019; Khodak et al., 2019) and showed guarantees for convex losses by using tools from online convex optimization. Lastly, we remark that there are analyses for other representation learning schemes (Arora et al., 2019; McNamara and Balcan, 2017; Galanti et al., 2016; Alquier et al., 2016; Denevi et al., 2018).

## 3 NOTATION AND SETUP

**Notation.** Let $[n] = \{1, 2, \ldots, n\}$. We use $\|\cdot\|$ or $\|\cdot\|_2$ to denote the $\ell_2$ norm of a vector or the spectral norm of a matrix. Denote by $\|\cdot\|_F$ and $\|\cdot\|_*$ the Frobenius norm and the nuclear norm of a matrix, respectively. Let $\langle \cdot, \cdot \rangle$ be the Euclidean inner product between vectors or matrices. Denote by $I$ the identity matrix. Let $\mathcal{N}(\mu, \sigma^2)/\mathcal{N}(\boldsymbol{\mu}, \Sigma)$ be the one-dimensional/multi-dimensional Gaussian distribution, and $\chi^2(m)$ the chi-squared distribution with $m$ degrees of freedom.

For a matrix $A \in \mathbb{R}^{m \times n}$, let $\sigma_i(A)$ be its $i$-th largest singular value. Let $\mathrm{span}(A)$ be the subspace of $\mathbb{R}^m$ spanned by the columns of $A$, i.e., $\mathrm{span}(A) = \{Av \mid v \in \mathbb{R}^n\}$. Denote $P_A = A(A^\top A)^\dagger A^\top \in \mathbb{R}^{m \times m}$, which is the projection matrix onto $\mathrm{span}(A)$. Here $\dagger$ stands for the Moore-Penrose pseudo-inverse. Note that $0 \preceq P_A \preceq I$ and $P_A^2 = P_A$. We also define $P_A^\perp = I - P_A$, which is the projection matrix onto $\mathrm{span}(A)^\perp$, the orthogonal complement of $\mathrm{span}(A)$ in $\mathbb{R}^m$. For a positive semidefinite (psd) matrix $B$, denote by $\lambda_{\max}(B)$ and $\lambda_{\min}(B)$ its largest and smallest eigenvalues, respectively; let $B^{1/2}$ be the psd matrix such that $(B^{1/2})^2 = B$.

We use the standard $O(\cdot)$, $\Omega(\cdot)$ and $\Theta(\cdot)$ notation to hide universal constant factors. We also use $a \lesssim b$ or $b \gtrsim a$ to indicate $a = O(b)$, and use $a \gg b$ or $b \ll a$ to mean that $a \geq C \cdot b$ for a sufficiently large universal constant $C > 0$.

**Problem Setup.** Suppose that there are $T$ source tasks. Each task $t \in [T]$ is associated with a distribution $\mu_t$ over the joint data space $\mathsf{X} \times \mathsf{Y}$, where $\mathsf{X}$ is the input space and $\mathsf{Y}$ is the output space. In this paper we consider $\mathsf{X} \subseteq \mathbb{R}^d$ and $\mathsf{Y} \subseteq \mathbb{R}$. For each source task $t \in [T]$ we have access to $n_1$ i.i.d. samples $(\boldsymbol{x}_{t,1}, y_{t,1}), \ldots, (\boldsymbol{x}_{t,n_1}, y_{t,n_1})$ from $\mu_t$. For convenience, we express these $n_1$ samples collectively as an input matrix $X_t \in \mathbb{R}^{n_1 \times d}$ and an output vector $\boldsymbol{y}_t \in \mathbb{R}^{n_1}$.

Multitask learning tries to learn prediction functions for all the $T$ source tasks simultaneously in the hope of discovering some underlying common property of these tasks. The common property we consider in this paper is a *representation*, which is a function $\phi : \mathsf{X} \to \mathsf{Z}$ that maps an input to some feature space $\mathsf{Z} \subseteq \mathbb{R}^k$. We restrict the representation function to be in some function class $\Phi$, e.g., neural networks. We try to use different linear predictors on top of a common representation function $\phi$ to model the input-output relations in different source tasks. Namely, for each task $t \in [T]$, we set the prediction function to be $\boldsymbol{x} \mapsto \langle \boldsymbol{w}_t, \phi(\boldsymbol{x}) \rangle$ $(\boldsymbol{w}_t \in \mathbb{R}^k)$. Therefore, using the training samples from $T$ tasks, we can solve the following optimization problem to learn the representation:[2]

$$\underset{\phi \in \Phi, \boldsymbol{w}_1, \ldots, \boldsymbol{w}_T \in \mathbb{R}^k}{\text{minimize}} \frac{1}{2n_1 T} \sum_{t=1}^{T} \sum_{i=1}^{n_1} \left( y_{t,i} - \langle \boldsymbol{w}_t, \phi(\boldsymbol{x}_{t,i}) \rangle \right)^2. \tag{1}$$

We overload the notation to allow $\phi$ to apply to all the samples in a data matrix simultaneously, i.e., $\phi(X_t) = [\phi(\boldsymbol{x}_{t,1}), \ldots, \phi(\boldsymbol{x}_{t,n_1})]^\top \in \mathbb{R}^{n_1 \times k}$. Then (1) can be rewritten as

$$\underset{\phi \in \Phi, \boldsymbol{w}_1, \ldots, \boldsymbol{w}_T \in \mathbb{R}^k}{\text{minimize}} \frac{1}{2n_1 T} \sum_{t=1}^{T} \left\| \boldsymbol{y}_t - \phi(X_t) \boldsymbol{w}_t \right\|^2. \tag{2}$$

---

[2] We use the $\ell_2$ loss throughout this paper.

Let $\hat{\phi} \in \Phi$ be the representation function obtained by solving (2). Now we retain this representation and apply it to future (target) tasks. For a target task specified by a distribution $\mu_{T+1}$ over $\mathsf{X} \times \mathsf{Y}$, suppose we receive $n_2$ i.i.d. samples $X_{T+1} \in \mathbb{R}^{n_2 \times d}, \boldsymbol{y}_{T+1} \in \mathbb{R}^{n_2}$. We further train a linear predictor on top of $\hat{\phi}$ for this task:

$$\underset{\boldsymbol{w}_{T+1} \in \mathbb{R}^k}{\text{minimize}} \frac{1}{2n_2} \left\| \boldsymbol{y}_{T+1} - \hat{\phi}(X_{T+1})\boldsymbol{w}_{T+1} \right\|^2. \tag{3}$$

Let $\hat{\boldsymbol{w}}_{T+1}$ be the returned solution. We are interested in whether our learned predictor $\boldsymbol{x} \mapsto \langle \hat{\boldsymbol{w}}_{T+1}, \hat{\phi}(\boldsymbol{x}) \rangle$ works well on average for the target task, i.e., we want the population loss

$$L_{\mu_{T+1}}(\hat{\phi}, \hat{\boldsymbol{w}}_{T+1}) = \mathbb{E}_{(\boldsymbol{x},y) \sim \mu_{T+1}} \frac{1}{2}(y - \langle \hat{\boldsymbol{w}}_{T+1}, \hat{\phi}(\boldsymbol{x}) \rangle)^2$$

to be small. In particular, we are interested in the *few-shot* learning setting, where the number of samples $n_2$ from the target task is small – much smaller than the number of samples required for learning the target task from scratch.

**Data assumption.** In order for the above learning procedure to make sense, we assume that there is a ground-truth optimal representation function $\phi^* \in \Phi$ and specializations $\boldsymbol{w}_1^*, \ldots, \boldsymbol{w}_{T+1}^* \in \mathbb{R}^k$ for all the tasks such that for each task $t \in [T+1]$, we have $\mathbb{E}_{(\boldsymbol{x},y) \sim \mu_t}[y|\boldsymbol{x}] = \langle \boldsymbol{w}_t^*, \phi^*(\boldsymbol{x}) \rangle$. More specifically, we assume $(\boldsymbol{x}, y) \sim \mu_t$ can be generated by

$$y = \langle \boldsymbol{w}_t^*, \phi^*(\boldsymbol{x}) \rangle + z, \quad \boldsymbol{x} \sim p_t, z \sim \mathcal{N}(0, \sigma^2), \tag{4}$$

where $\boldsymbol{x}$ and $z$ are independent. Our goal is to bound the *excess risk* of our learned model on the target task, i.e., how much our learned model $(\hat{\phi}, \hat{\boldsymbol{w}}_{T+1})$ performs worse than the optimal model $(\phi^*, \boldsymbol{w}_{T+1}^*)$ on the target task:

$$\begin{aligned} \text{ER}(\hat{\phi}, \hat{\boldsymbol{w}}_{T+1}) &= L_{\mu_{T+1}}(\hat{\phi}, \hat{\boldsymbol{w}}_{T+1}) - L_{\mu_{T+1}}(\phi^*, \boldsymbol{w}_{T+1}^*) \\ &= \frac{1}{2} \mathbb{E}_{\boldsymbol{x} \sim p_{T+1}}[(\langle \hat{\boldsymbol{w}}_{T+1}, \hat{\phi}(\boldsymbol{x}) \rangle - \langle \boldsymbol{w}_{T+1}^*, \phi^*(\boldsymbol{x}) \rangle)^2]. \end{aligned} \tag{5}$$

Here we have used the relation (4). Oftentimes we are interested in the average performance on a random target task (i.e., $\boldsymbol{w}_{T+1}^*$ is random). In such case we look at the expected excess risk $\mathbb{E}_{\boldsymbol{w}_{T+1}^*}[\text{ER}(\hat{\phi}, \hat{\boldsymbol{w}}_{T+1})]$.

## 4 LOW-DIMENSIONAL LINEAR REPRESENTATIONS

In this section, we consider the case where the representation is a linear map from the original input space $\mathbb{R}^d$ to a low-dimensional space $\mathbb{R}^k$ ($k \ll d$). Namely, we let the representation function class be $\Phi = \{\boldsymbol{x} \mapsto B^\top \boldsymbol{x} \mid B \in \mathbb{R}^{d \times k}\}$. Then the optimization problem (2) for learning the representation can be written as:

$$(\hat{B}, \hat{W}) \leftarrow \underset{\substack{B \in \mathbb{R}^{d \times k} \\ W = [\boldsymbol{w}_1, \ldots, \boldsymbol{w}_T] \in \mathbb{R}^{k \times T}}}{\arg\min} \frac{1}{2n_1 T} \sum_{t=1}^T \|\boldsymbol{y}_t - X_t B \boldsymbol{w}_t\|^2. \tag{6}$$

The inputs from $T$ source tasks, $X_1, \ldots, X_T \in \mathbb{R}^{n_1 \times d}$, can be written in the form of a linear operator $\mathcal{X} : \mathbb{R}^{d \times T} \to \mathbb{R}^{n_1 \times T}$, where

$$\mathcal{X}(\Theta) = [X_1 \boldsymbol{\theta}_1, \ldots, X_T \boldsymbol{\theta}_T], \quad \forall \Theta = [\boldsymbol{\theta}_1, \ldots, \boldsymbol{\theta}_T] \in \mathbb{R}^{d \times T}.$$

With this notation, (6) can be rewritten as

$$(\hat{B}, \hat{W}) \leftarrow \underset{B \in \mathbb{R}^{d \times k}, W \in \mathbb{R}^{k \times T}}{\arg\min} \frac{1}{2n_1 T} \|Y - \mathcal{X}(BW)\|_F^2, \text{ where } Y = [\boldsymbol{y}_1, \ldots, \boldsymbol{y}_T] \in \mathbb{R}^{n_1 \times T}. \tag{7}$$

With the learned representation $\hat{B}$ from (7), for the target task, we further find a linear function on top of the representation:

$$\hat{\boldsymbol{w}}_{T+1} \leftarrow \underset{\boldsymbol{w} \in \mathbb{R}^k}{\arg\min} \frac{1}{2n_2} \left\| \boldsymbol{y}_{T+1} - X_{T+1} \hat{B} \boldsymbol{w} \right\|^2. \tag{8}$$

As described in Section 3, we assume that all $T+1$ tasks share a common ground-truth representation specified by a matrix $B^* \in \mathbb{R}^{d \times k}$ such that a sample $(\boldsymbol{x}, y) \sim \mu_t$ satisfies $\boldsymbol{x} \sim p_t$ and $y = (\boldsymbol{w}_t^*)^\top (B^*)^\top \boldsymbol{x} + z$ where $z \sim \mathcal{N}(0, \sigma^2)$ is independent of $\boldsymbol{x}$. Here $\boldsymbol{w}_t^* \in \mathbb{R}^k$, and we assume $\|\boldsymbol{w}_t^*\| = \Theta(1)$ for all $t \in [T+1]$. Denote $W^* = [\boldsymbol{w}_1^*, \dots, \boldsymbol{w}_T^*] \in \mathbb{R}^{k \times T}$. Then we can write $Y = \mathcal{X}(B^*W^*) + Z$, where the noise matrix $Z$ has i.i.d. $\mathcal{N}(0, \sigma^2)$ entries.

Assume $\mathbb{E}_{\boldsymbol{x} \sim p_t}[\boldsymbol{x}] = \boldsymbol{0}$ and let $\Sigma_t = \mathbb{E}_{\boldsymbol{x} \sim p_t}[\boldsymbol{x}\boldsymbol{x}^\top]$ for all $t \in [T+1]$. Note that a sample $\boldsymbol{x} \sim p_t$ can be generated from $\boldsymbol{x} = \Sigma_t^{1/2} \bar{\boldsymbol{x}}$ for $\bar{\boldsymbol{x}} \sim \bar{p}_t$ such that $\mathbb{E}_{\bar{\boldsymbol{x}} \sim \bar{p}_t}[\bar{\boldsymbol{x}}] = \boldsymbol{0}$ and $\mathbb{E}_{\bar{\boldsymbol{x}} \sim \bar{p}_t}[\bar{\boldsymbol{x}}\bar{\boldsymbol{x}}^\top] = I$. ($\bar{p}_t$ is called the whitening of $p_t$.) In this section we make the following assumptions on the input distributions $p_1, \dots, p_{T+1}$.

**Assumption 4.1** (subgaussian input). *There exists $\rho > 0$ such that, for all $t \in [T+1]$, the random vector $\bar{\boldsymbol{x}} \sim \bar{p}_t$ is $\rho^2$-subgaussian.*[3]

**Assumption 4.2** (covariance dominance). *There exists $c > 0$ such that $\Sigma_t \succeq c \cdot \Sigma_{T+1}$ for all $t \in [T]$.*[4]

Assumption 4.1 is a standard assumption in statistical learning to obtain probabilistic tail bounds used in our proof. It may be replaced with other moment or boundedness conditions if we adopt different tail bounds in the analysis.

Assumption 4.2 says that every direction spanned by $\Sigma_{T+1}$ should also be spanned by $\Sigma_t$ ($t \in [T]$), and the parameter $c$ quantifies how "easy" it is for $\Sigma_t$ to cover $\Sigma_{T+1}$. Intuitively, the larger $c$ is, the easier it is to cover the target domain using source domains, and we will indeed see that the risk will be proportional to $\frac{1}{c}$. We remark that we do not necessarily need $\Sigma_t \succeq c \cdot \Sigma_{T+1}$ for all $t \in [T]$; as long as this holds for a constant fraction of $t$'s, our result is valid.

We also make the following assumption that characterizes the diversity of the source tasks.

**Assumption 4.3** (diverse source tasks). *The matrix $W^* = [\boldsymbol{w}_1^*, \dots, \boldsymbol{w}_T^*] \in \mathbb{R}^{k \times T}$ satisfies $\sigma_k^2(W^*) \geq \Omega(\frac{T}{k})$.*

Recall that $\|\boldsymbol{w}_t^*\| = \Theta(1)$, which implies $\sum_{j=1}^k \sigma_j^2(W^*) = \|W^*\|_F^2 = \Theta(T)$. Thus, Assumption 4.3 is equivalent to saying that $\frac{\sigma_1(W^*)}{\sigma_k(W^*)} = O(1)$. Roughly speaking, this means that $\{\boldsymbol{w}_t^*\}_{t \in [T]}$ can cover all directions in $\mathbb{R}^k$. As an example, Assumption 4.3 is satisfied with high probability when $\boldsymbol{w}_t^*$'s are sampled i.i.d. from $\mathcal{N}(0, \Sigma)$ with $\frac{\lambda_{\max}(\Sigma)}{\lambda_{\min}(\Sigma)} = O(1)$.

Finally, we make the following assumption on the distribution of the target task.

**Assumption 4.4** (distribution of target task). *Assume that $\boldsymbol{w}_{T+1}^*$ follows a distribution $\nu$ such that $\left\| \mathbb{E}_{\boldsymbol{w} \sim \nu}[\boldsymbol{w}\boldsymbol{w}^\top] \right\| \leq O\left(\frac{1}{k}\right)$.*

Since we assume $\|\boldsymbol{w}_{T+1}^*\| = \Theta(1)$, the assumption $\left\| \mathbb{E}_{\boldsymbol{w} \sim \nu}[\boldsymbol{w}\boldsymbol{w}^\top] \right\| \leq O(\frac{1}{k})$ means that the distribution of $\boldsymbol{w}_{T+1}^*$ does not align with any direction significantly more than average. It is useful to think of the uniform distribution on the unit sphere as an example, though we can allow a much more general class of distributions. This is also compatible with Assumption 4.3 which says that $\boldsymbol{w}_t^*$'s cover all the directions.

Assumption 4.4 can be removed at the cost of a slightly worse risk bound. See Remark 4.1. Our main result in this section is the following theorem.

**Theorem 4.1** (main theorem for linear representations). *Fix a failure probability $\delta \in (0, 1)$. Under Assumptions 4.1, 4.2, 4.3 and 4.4, we further assume $2k \leq \min\{d, T\}$ and that the sample sizes in source and target tasks satisfy $n_1 \gg \rho^4(d + \log \frac{T}{\delta})$, $n_2 \gg \rho^4(k + \log \frac{1}{\delta})$, and $cn_1 \geq n_2$. Define $\kappa = \frac{\max_{t \in [T]} \lambda_{\max}(\Sigma_t)}{\min_{t \in [T]} \lambda_{\min}(\Sigma_t)}$. Then with probability at least $1 - \delta$ over the samples, the expected excess risk of the learned predictor $\boldsymbol{x} \mapsto \hat{\boldsymbol{w}}_{T+1}^\top \hat{B} \boldsymbol{x}$ on the target task satisfies*

$$\mathbb{E}_{\boldsymbol{w}_{T+1}^* \sim \nu}[\mathrm{ER}(\hat{B}, \hat{\boldsymbol{w}}_{T+1})] \lesssim \sigma^2 \left( \frac{kd \log(\kappa n_1)}{cn_1 T} + \frac{k + \log \frac{1}{\delta}}{n_2} \right). \tag{9}$$

---

[3]A random vector $\boldsymbol{x}$ is called $\rho^2$-subgaussian if for any fixed unit vector $\boldsymbol{v}$ of the same dimension, the random variable $\boldsymbol{v}^\top \boldsymbol{x}$ is $\rho^2$-subgaussian, i.e., $\mathbb{E}[e^{s \cdot \boldsymbol{v}^\top (\boldsymbol{x} - \mathbb{E}[\boldsymbol{x}])}] \leq e^{s^2 \rho^2 / 2}$ ($\forall s \in \mathbb{R}$).

[4]Note that Assumption 4.2 is a significant generalization of the identically distributed isotropic assumption used in concurrent work Tripuraneni et al. (2020): they require $\Sigma_1 = \Sigma_2 = \dots = \Sigma_{T+1} = I$.

The proof of Theorem 4.1 is in Appendix A. Theorem 4.1 shows that it is possible to learn the target task using only $O(k)$ samples via learning a good representation from the source tasks, which is better than the baseline $O(d)$ sample complexity for linear regression, thus demonstrating the benefit of representation learning. It also shows that all $n_1 T$ samples from source tasks can be pooled together, bypassing the $\Omega(\frac{1}{T})$ barrier under the i.i.d. tasks assumption.

**Remark 4.1** (deterministic target task). *We can drop Assumption 4.4 and easily obtain the following excess risk bound for any deterministic $\boldsymbol{w}_{T+1}^*$ by slightly modifying the proof of Theorem 4.1:*

$$\mathrm{ER}(\hat{B}, \hat{\boldsymbol{w}}_{T+1}) \lesssim \sigma^2 \left( \frac{k^2 d \log(\kappa n_1)}{c n_1 T} + \frac{k^2}{c n_1} + \frac{k + \log \frac{1}{\delta}}{n_2} \right),$$

*which is only at most $k$ times larger than the bound in* (9).

## 5 GENERAL LOW-DIMENSIONAL REPRESENTATIONS

Now we return to the general case described in Section 3 where we allow a general representation function class $\Phi$. We still assume that the representation is of low dimension $k$ like in Section 4, and we assume that inputs from all the tasks follow the same distribution, i.e., $p_1 = \cdots = p_{T+1} = p$, but each task $t$ still has its own specialization function $\boldsymbol{w}_t^*$ (c.f. (4)). We overload the notation from Section 4 and use $\mathcal{X}$ to represent the collection of all the training inputs from $T$ source tasks $X_1, \ldots, X_T \in \mathbb{R}^{n_1 \times d}$. We can think of $\mathcal{X}$ as a third-order tensor of dimension $n_1 \times d \times T$.

To characterize the complexity of the representation function class $\Phi$, we need the standard definition of Gaussian width.

**Definition 5.1** (Gaussian width). *Given a set $\mathcal{K} \subset \mathbb{R}^m$, the Gaussian width of $\mathcal{K}$ is defined as:* $\mathcal{G}(\mathcal{K}) := \mathbb{E}_{\boldsymbol{z} \sim \mathcal{N}(\boldsymbol{0}, I)} \sup_{\boldsymbol{v} \in \mathcal{K}} \langle \boldsymbol{v}, \boldsymbol{z} \rangle$.

We will measure the complexity of $\Phi$ using the Gaussian width of the following set that depends on the input data $\mathcal{X}$:

$$\begin{aligned} \mathcal{F}_{\mathcal{X}}(\Phi) = \big\{ A = [\boldsymbol{a}_1, \ldots, \boldsymbol{a}_T] \in \mathbb{R}^{n_1 \times T} : \|A\|_F = 1, \\ \exists \phi, \phi' \in \Phi \text{ s.t. } \boldsymbol{a}_t \in \mathrm{span}([\phi(X_t), \phi'(X_t)]), \forall t \in [T] \big\}. \end{aligned} \tag{10}$$

We also need the following definition.

**Definition 5.2** (covariance between two representations). *Given a distribution $q$ over $\mathbb{R}^d$ and two representation functions $\phi, \phi' \in \Phi$, define the covariance between $\phi$ and $\phi'$ with respect to $q$ to be*

$$\Sigma_q(\phi, \phi') = \mathbb{E}_{\boldsymbol{x} \sim q} \big[ \phi(\boldsymbol{x}) \phi'(\boldsymbol{x})^\top \big] \in \mathbb{R}^{k \times k}.$$

*Also define the symmetric covariance as*

$$\Lambda_q(\phi, \phi') = \begin{bmatrix} \Sigma_q(\phi, \phi) & \Sigma_q(\phi, \phi') \\ \Sigma_q(\phi', \phi) & \Sigma_q(\phi', \phi') \end{bmatrix} \in \mathbb{R}^{2k \times 2k}.$$

It is easy to verify $\Lambda_q(\phi, \phi') \succeq 0$ for any $\phi, \phi'$ and $q$, as shown in the proof of Lemma B.2.

We make the following assumptions on the input distribution $p$, which ensure concentration properties of the representation covariances.

**Assumption 5.1** (point-wise concentration of covariance). *For $\delta \in (0, 1)$, there exists a number $N_{\mathrm{point}}(\Phi, p, \delta)$ such that if $n \geq N_{\mathrm{point}}(\Phi, p, \delta)$, then for any given $\phi, \phi' \in \Phi$, $n$ i.i.d. samples of $p$ will with probability at least $1 - \delta$ satisfy*

$$0.9 \Lambda_p(\phi, \phi') \preceq \Lambda_{\hat{p}}(\phi, \phi') \preceq 1.1 \Lambda_p(\phi, \phi'),$$

*where $\hat{p}$ is the empirical distribution over the $n$ samples.*

**Assumption 5.2** (uniform concentration of covariance). *For $\delta \in (0, 1)$, there exists a number $N_{\mathrm{unif}}(\Phi, p, \delta)$ such that if $n \geq N_{\mathrm{unif}}(\Phi, p, \delta)$, then $n$ i.i.d. samples of $p$ will with probability at least $1 - \delta$ satisfy*

$$0.9 \Lambda_p(\phi, \phi') \preceq \Lambda_{\hat{p}}(\phi, \phi') \preceq 1.1 \Lambda_p(\phi, \phi'), \quad \forall \phi, \phi' \in \Phi,$$

*where $\hat{p}$ is the empirical distribution over the $n$ samples.*

Assumptions 5.1 and 5.2 are conditions on the representation function class $\Phi$ and the input distribution $p$ that ensure concentration of empirical covariances to their population counterparts. Typically, we expect $N_{\mathrm{unif}}(\Phi, p, \delta) \gg N_{\mathrm{point}}(\Phi, p, \delta)$ since uniform concentration is a stronger requirement. In Section 4, we have essentially shown that for linear representations and subgaussian input distributions, $N_{\mathrm{unif}}(\Phi, p, \delta) = \tilde{O}(d)$ and $N_{\mathrm{point}}(\Phi, p, \delta) = \tilde{O}(k)$ (see Claims A.1 and A.2).

Our main theorem in this section is the following:

**Theorem 5.1** (main theorem for general representations). *Fix a failure probability $\delta \in (0, 1)$. Suppose $n_1 \geq N_{\mathrm{unif}}\left(\Phi, p, \frac{\delta}{3T}\right)$ and $n_2 \geq N_{\mathrm{point}}\left(\Phi, p, \frac{\delta}{3}\right)$. Under Assumptions 4.3 and 4.4, with probability at least $1 - \delta$ over the samples, the expected excess risk of the learned predictor $\boldsymbol{x} \mapsto \hat{\boldsymbol{w}}_{T+1}^\top \hat{\phi}(\boldsymbol{x})$ on the target task satisfies*

$$\mathbb{E}_{\boldsymbol{w}_{T+1}^* \sim \nu}[\mathrm{ER}(\hat{\phi}, \hat{\boldsymbol{w}}_{T+1})] \lesssim \sigma^2 \left( \frac{\mathcal{G}(\mathcal{F}_{\mathcal{X}}(\Phi))^2 + \log \frac{1}{\delta}}{n_1 T} + \frac{k + \log \frac{1}{\delta}}{n_2} \right). \tag{11}$$

Theorem 5.1 is very similar to Theorem 4.1 in terms of the result and the assumptions made. In the bound (11), the complexity of $\Phi$ is captured by the Gaussian width of the data-dependent set $\mathcal{F}_{\mathcal{X}}(\Phi)$ defined in (10). Data-dependent complexity measures are ubiquitous in generalization theory, one of the most notable examples being Rademacher complexity. Similar complexity measure also appeared in existing representation learning theory (Maurer et al., 2016). Usually, for specific examples, we can apply concentration bounds to get rid of the data dependency, such as our result for linear representations (Theorem 4.1).

Our assumptions on the linear specification functions $\boldsymbol{w}_t^*$'s are the same as in Theorem 4.1. The probabilistic assumption on $\boldsymbol{w}_{T+1}^*$ can also be removed at the cost of an additional factor of $k$ in the bound – see Remark 4.1. We defer the full proof of Theorem 5.1 to Appendix B.

# 6 HIGH-DIMENSIONAL LINEAR REPRESENTATIONS

In this section, we consider the case where the representation is a general linear map without an explicit dimensionality constraint, and we will prove a norm-based result by exploiting the *intrinsic dimension* of the representation. Such a generalization is desirable since in many applications the representation dimension is not restricted.

Without loss of generality, we let the representation function class be $\Phi = \{\boldsymbol{x} \mapsto B^\top \boldsymbol{x} \mid B \in \mathbb{R}^{d \times T}\}$. We note that a dimension-$T$ representation is sufficient for learning $T$ source tasks and any choice of dimension greater than $T$ will not change our argument. We use the same notation from Section 4 unless otherwise specified.

In this section we additionally assume that all tasks have the same input covariance:

**Assumption 6.1.** *The input distributions in all tasks satisfy $\Sigma_1 = \cdots = \Sigma_{T+1} = \Sigma$.*

Note that each task $t$ still has its own specialization function $\boldsymbol{w}_t^*$ (c.f. (4)). We remark that there are many interesting and nontrivial scenarios under Assumption 6.1 – for example, consider the case where the inputs in each task are all images from ImageNet and each task asks whether the image is from a specific class.

Since we do not have a dimensionality constraint, we modify (7) by adding norm constraints:

$$(\hat{B}, \hat{W}) \leftarrow \underset{B \in \mathbb{R}^{d \times T}, W \in \mathbb{R}^{T \times T}}{\arg\min} \frac{1}{2n_1} \|Y - \mathcal{X}(BW)\|_F^2 + \frac{\lambda}{2} \|W\|_F^2 + \frac{\lambda}{2} \|B\|^2. \tag{12}$$

For the target task, we also modify (8) by adding a norm constraint:

$$\hat{\boldsymbol{w}}_{T+1} \leftarrow \underset{\|\boldsymbol{w}\| \leq r}{\arg\min} \frac{1}{2n_2} \|X_{T+1} \hat{B} \boldsymbol{w} - \boldsymbol{y}_{T+1}\|^2. \tag{13}$$

We will specify the choices of regularization, i.e., $\lambda$ and $r$ in Theorem 6.1.

Similar to Section 4, the source task data relation is denoted as $Y = \mathcal{X}(\Theta^*) + Z$, where $\Theta^* \in \mathbb{R}^{d \times T}$ is the ground truth and $Z$ has i.i.d. $\mathcal{N}(0, \sigma^2)$ entries. Suppose that the target task data satisfy $\boldsymbol{y}_{T+1} = X_{T+1} \boldsymbol{\theta}_{T+1}^* + \boldsymbol{z}_{T+1} \in \mathbb{R}^{n_2}$. Similar to the setting in Section 4, we assume the target task data is subgaussian as in Assumption 4.1.

**Theorem 6.1** (main theorem for high-dimensional representations). *Fix a failure probability $\delta \in (0, 1)$. Under Assumptions 4.1 and 6.1, we further assume $n_1 \geq n_2$, $R = \|\Theta\|_*$. Let $r = 2\sqrt{R/T}$, $\bar{R} = R/\sqrt{T}$ and proper $\lambda$ specified in Lemma C.2. Let the target task model $\boldsymbol{\theta}_{T+1}^*$ be coherent with the source task models $\Theta^*$ in the sense that $\boldsymbol{\theta}_{T+1}^* \sim \nu = \mathcal{N}(\mathbf{0}, \Theta^*(\Theta^*)^\top/T)$. Then with probability at least $1 - \delta$ over the samples, the expected excess risk of the learned predictor $\boldsymbol{x} \mapsto \hat{\boldsymbol{w}}_{T+1}^\top \hat{B}^\top \boldsymbol{x}$ on the target task satisfies:*

$$\mathbb{E}_{\boldsymbol{\theta}_{T+1}^* \sim \nu}[\mathrm{ER}(\hat{B}, \hat{\boldsymbol{w}}_{T+1})] \leq \sigma\bar{R} \cdot \tilde{O}\left(\frac{\sqrt{\mathrm{Tr}(\Sigma)}}{\sqrt{n_1 T}} + \frac{\sqrt{\|\Sigma\|_2}}{\sqrt{n_2}}\right) + \zeta_{n_1, n_2}, \tag{14}$$

*where $\zeta_{n_1,n_2} := \rho^4\bar{R}^2\tilde{O}\left(\frac{\mathrm{Tr}(\Sigma)}{n_1} + \frac{\|\Sigma\|}{n_2}\right)$ is lower-order terms due to randomness of the input data. Here $\tilde{O}$ hides logarithmic factors.*

The proof of Theorem 6.1 is given in Appendix C. Note that $\|\Theta^*\|_F = \sqrt{T}$ when each $\boldsymbol{\theta}_t^*$ is of unit norm. Thus $\bar{R} = \|\Theta^*\|_* / \sqrt{T}$ should generally be regarded as $O(1)$ for a well-behaved $\Theta^*$ that is nearly low-dimensional. In this regime, Theorem 6.1 indicates that we are able to exploit all $n_1 T$ samples from the source tasks, similar to Theorem 4.1.

With a good representation, the sample complexity on the target task can also improve over learning the target task from scratch. Consider the baseline of regular ridge regression directly applied to the target task data:

$$\hat{\boldsymbol{\theta}} \leftarrow \underset{\|\boldsymbol{\theta}\| \leq \|\boldsymbol{\theta}_{T+1}^*\|}{\arg\min} \frac{1}{2n_2}\|X_{T+1}\boldsymbol{\theta} - \boldsymbol{y}_{T+1}\|^2. \tag{15}$$

Its standard excess risk bound in fixed design is $\mathrm{ER}(\hat{\boldsymbol{\theta}}^\lambda) \lesssim \sigma\sqrt{\frac{\|\boldsymbol{\theta}_{T+1}^*\|_2^2 \mathrm{Tr}(\Sigma)}{n_2}}$. (See e.g. Hsu et al. (2012).) Taking expectation over $\boldsymbol{\theta}_{T+1}^* \sim \nu = \mathcal{N}(\mathbf{0}, \Theta^*(\Theta^*)^\top/T)$, we obtain

$$\mathbb{E}_{\boldsymbol{\theta}_{T+1}^* \sim \nu}[\mathrm{ER}(\hat{\boldsymbol{\theta}}^\lambda)] \lesssim \sigma\frac{\|\Theta^*\|_F}{\sqrt{T}}\sqrt{\frac{\mathrm{Tr}(\Sigma)}{n_2}}. \tag{16}$$

Compared with (16), our bound (14) is an improvement as long as $\frac{\|\Theta^*\|_*^2}{\|\Theta^*\|_F^2} \ll \frac{\mathrm{Tr}(\Sigma)}{\|\Sigma\|}$. The left hand side $\frac{\|\Theta^*\|_*^2}{\|\Theta^*\|_F^2}$ is always no more than the rank of $\Theta^*$, and we call it the *intrinsic rank*. Hence we see that we can gain from representation learning if the source predictors are intrinsically low dimensional.

To intuitively understand how this is achieved, we note that a representation $B$ is reweighing linear combinations of the features according to their "importance" on the $T$ source tasks. We make an analogy with a simple case of feature selection. Suppose we have learned a representation vector $\boldsymbol{b}$ where $b_i$ scales with the importance of the $i$-th feature, i.e., the representation is $\phi(\boldsymbol{x}) = \boldsymbol{x} \odot \boldsymbol{b}$ (entry-wise product). Then ridge regression on the target task data $(X, \boldsymbol{y})$, $\mathrm{minimize}_{\|\boldsymbol{w}\| \leq r} \frac{1}{2n_2}\|X \cdot \mathrm{diag}(\boldsymbol{b}) \cdot \boldsymbol{w} - \boldsymbol{y}\|_2^2$, is equivalent to $\mathrm{minimize}_{\|\mathrm{diag}(\boldsymbol{b})^{-1}\boldsymbol{v}\| \leq r} \frac{1}{2n_2}\|X\boldsymbol{v} - \boldsymbol{y}\|_2^2$. From the above equation, we see that the features with large $|b_i|$ (those that were useful on the source tasks) will be more heavily used than the ones with small $|b_i|$ due to the reweighed $\ell_2$ constraint. Thus the important features are learned from the source tasks, and the coefficients are learned from the target task.

**Remark 6.1** (The non-convex landscape). *Although the optimization problem (12) is non-convex, its structure allows us to apply existing landscape analysis of matrix factorization problems (Haeffele et al., 2014) and to show that it has the nice properties of no strict saddles and no bad local minima. Therefore, randomly initialized gradient descent or perturbed gradient descent are guaranteed to converge to a global minimum of (12) (Ge et al., 2015; Lee et al., 2016; Jin et al., 2017).*

**Remark 6.2** (Multi-class problems). *When both source and target have multi-class labels instead of independent tasks, using quadratic loss on the one-hot labels, our results apply similarly and will attain an excess risk of the form $\sigma\bar{R}\tilde{O}\left(\frac{\sqrt{\mathrm{Tr}(\Sigma)}}{\sqrt{n_1}} + \frac{\sqrt{\|\Sigma\|_2}}{\sqrt{n_2}}\right)$ plus lower-order terms (see e.g. Lee et al. (2020)). Notice the result is independent of the number of classes.*

## 7 NEURAL NETWORKS

In this section, we show that we can provably learn good representations in a neural network.

Consider a two-layer ReLU neural network $f_{B,\boldsymbol{w}}(\boldsymbol{x}) = \boldsymbol{w}^\top(B^\top\boldsymbol{x})_+$, where $\boldsymbol{w} \in \mathbb{R}^d$, $B \in \mathbb{R}^{d_0 \times d}$ and $\boldsymbol{x} \in \mathbb{R}^{d_0}$. Here $(\cdot)_+$ is the ReLU activation $(z)_+ = \max\{0, z\}$ defined element-wise. Namely, we let the representation function class be $\Phi = \{\boldsymbol{x} \to (B^\top\boldsymbol{x})_+ | B \in \mathbb{R}^{d_0 \times d}\}$. On the source tasks we use the square loss with weight decay regularizer:[5]

$$(\hat{B}, \hat{W}) \leftarrow \underset{B \in \mathbb{R}^{d_0 \times d}, W = [\boldsymbol{w}_1, \cdots \boldsymbol{w}_T] \in \mathbb{R}^{d \times T}}{\arg\min} \frac{1}{2n_1 T} \sum_{t=1}^T \|\boldsymbol{y}_t - (X_t B)_+ \boldsymbol{w}_t\|^2 + \frac{\lambda}{2}\|B\|_F^2 + \frac{\lambda}{2}\|W\|_F^2. \tag{17}$$

On the target task, we simply re-train the output layer while fixing the hidden layer weights:

$$\hat{\boldsymbol{w}}_{T+1} \leftarrow \underset{\|\boldsymbol{w}\| \leq r}{\arg\min} \frac{1}{2n_2}\|\boldsymbol{y}_{T+1} - (X_{T+1}\hat{B})_+ \boldsymbol{w}\|^2. \tag{18}$$

**Assumption 7.1.** *All tasks share the same input distribution: $p_1 = \cdots = p_{T+1} = p$. We redefine $\Sigma$ to be the covariance operator of the feature induced by ReLU, i.e., it is a kernel defined by $\Sigma(\boldsymbol{u}, \boldsymbol{v}) = \mathbb{E}_{\boldsymbol{x} \sim p}[(\boldsymbol{u}^\top\boldsymbol{x})_+(\boldsymbol{v}^\top\boldsymbol{x})_+]$, for $\boldsymbol{u}, \boldsymbol{v}$ on the unit sphere $\mathbb{S}^{d_0-1} \subset \mathbb{R}^{d_0}$.*

**Assumption 7.2** (teacher network). *Assume for the source tasks that $\boldsymbol{y}_t = (X_t B^*)_+ \boldsymbol{w}_t^* + \boldsymbol{z}_t$ is generated by a teacher network with parameters $B^* \in \mathbb{R}^{d_0 \times d}$, $W^* = [\boldsymbol{w}_1^*, \cdots, \boldsymbol{w}_T^*] \in \mathbb{R}^{d \times T}$, and noise term $\boldsymbol{z}_t \sim \mathcal{N}(0, \sigma^2 I)$. A standard lifting of the neural network is: $f_{\alpha_t} = \langle \alpha_t, \phi(\boldsymbol{x}) \rangle$ where $\phi(\boldsymbol{x}) : \mathbb{S}^{d_0-1} \to \mathbb{R}, \phi(\boldsymbol{x})_{\boldsymbol{b}} = (\boldsymbol{b}^\top\boldsymbol{x})_+$ is the feature map, i.e., for each task, $\alpha_t(\boldsymbol{b}_i/\|\boldsymbol{b}_i\|) = W_{i,t}\|\boldsymbol{b}_i\|$ and is zero elsewhere. We assume $\alpha_{T+1}$ that describes the target function to follow a Gaussian process $\mu$ with covariance function $K(\boldsymbol{b}, \boldsymbol{b}') = \sum_{t=1}^T \alpha_t(\boldsymbol{b})\alpha_t(\boldsymbol{b}')$.*

**Theorem 7.1.** *Fix a failure probability $\delta \in (0, 1)$. Under Assumptions 4.1, 7.1 and 7.2, let $n_1 \geq n_2$, $\bar{R} = (\frac{1}{2}\|B^*\|_F^2 + \frac{1}{2}\|W^*\|_F^2)/\sqrt{T}$. Let the target task model $f_{\alpha_{T+1}} = \langle \alpha_{T+1}, \phi(\boldsymbol{x}) \rangle$ be coherent with the source task models in the sense that $\alpha_{T+1}^* \sim \nu$. Set $r^2 = (\|B^*\|_F^2 + \|W^*\|_F^2)/T$. Then with probability at least $1 - \delta$ over the samples, the expected excess risk of the learned predictor $\boldsymbol{x} \mapsto \hat{\boldsymbol{w}}_{T+1}^\top(\hat{B}^\top\boldsymbol{x})_+$ on the target task satisfies:*

$$\mathbb{E}_{\alpha_{T+1} \sim \nu}[\mathrm{ER}(f_{\hat{B}, \hat{\boldsymbol{w}}_{T+1}})] \leq \sigma\bar{R} \cdot \tilde{O}\left(\frac{\sqrt{\mathrm{Tr}(\Sigma)}}{\sqrt{n_1 T}} + \frac{\sqrt{\|\Sigma\|_2}}{\sqrt{n_2}}\right) + \zeta_{n_1, n_2}, \tag{19}$$

*where $\zeta_{n_1, n_2} := \rho^4 \bar{R}^2 \tilde{O}(\frac{\mathrm{Tr}(\Sigma)}{n_1} + \frac{\|\Sigma\|}{n_2})$ is lower-order term due to randomness of the input data.*

To highlight the advantage of representation learning, we compare to training a neural network with weight decay directly on the target task:

$$(\hat{B}, \hat{\boldsymbol{w}}) = \underset{B, \boldsymbol{w}, \|B\boldsymbol{w}\| \leq \bar{R}}{\arg\min} \frac{1}{2n} \sum_{i=1}^n \|\boldsymbol{y}_{t+1} - (X_{T+1}B)_+ \boldsymbol{w}\|^2. \tag{20}$$

The error of the baseline method in fixed-design is

$$\mathbb{E}[\mathrm{ER}(f_{\hat{B}, \hat{\boldsymbol{w}}})] \lesssim \sigma\bar{R}\sqrt{\frac{\mathrm{Tr}(\Sigma)}{n_2}}. \tag{21}$$

We see that Equation (19) is always smaller than Equation (21) since $n_1 T \geq n_2$. See Appendix D for the proof of Theorem 7.1 and the calculation of (21).

## 8 CONCLUSION

We gave the first statistical analysis showing that representation learning can fully exploit all data points from source tasks to enable few-shot learning on a target task. This type of results were shown for both low-dimensional and high-dimensional representation function classes.

There are many important directions to pursue in representation learning and few-shot learning. Our results in Sections 6 and 7 indicate that explicit low dimensionality is not necessary, and norm-based capacity control also forces the classifier to learn good representations. Further questions include whether this is a general phenomenon in all deep learning models, whether other capacity control can be applied, and how to optimize to attain good representations.

---

[5]Wei et al. (2019) show that (17) can be minimized in polynomial iteration complexity using perturbed gradient descent, though potentially exponential width is required.

## ACKNOWLEDGMENTS

SSD acknowledges support of National Science Foundation (Grant No. DMS-1638352) and the Infosys Membership. JDL acknowledges support of the ARO under MURI Award W911NF-11-1-0303, the Sloan Research Fellowship, and NSF CCF 2002272. WH is supported by NSF, ONR, Simons Foundation, Schmidt Foundation, Amazon Research, DARPA and SRC. QL is supported by NSF #2030859 and the Computing Research Association for the CIFellows Project. The authors also acknowledge the generous support of the Institute for Advanced Study on the Theoretical Machine Learning program, where SSD, WH, JDL, and QL were participants.

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

## A    PROOF OF THEOREM 4.1

We first prove several claims and then combine them to finish the proof of Theorem 4.1. We will use technical lemmas proved in Section A.1.

**Claim A.1** (covariance concentration of source tasks). *Suppose $n_1 \gg \rho^4(d + \log(T/\delta))$ for $\delta \in (0,1)$. Then with probability at least $1 - \frac{\delta}{10}$ over the inputs $X_1, \ldots, X_T$ in the source tasks, we have*

$$0.9\Sigma_t \preceq \frac{1}{n_1}X_t^\top X_t \preceq 1.1\Sigma_t, \quad \forall t \in [T]. \tag{22}$$

*Proof.* According to our assumption on $p_t$, we can write $X_t = \bar{X}_t \Sigma_t^{1/2}$, where $\bar{X}_t \in \mathbb{R}^{n_1 \times d}$ and the rows of $\bar{X}_t$ hold i.i.d. samples of $\bar{p}_t$. Since $\bar{p}_t$ satisfies the conditions in Lemma A.6, from Lemma A.6 we know that with probability at least $1 - \frac{\delta}{10T}$,

$$0.9I \preceq \frac{1}{n_1}\bar{X}_t^\top \bar{X}_t \preceq 1.1I,$$

which implies

$$0.9\Sigma_t \preceq \frac{1}{n_1}\Sigma_t^{1/2}\bar{X}_t^\top \bar{X}_t \Sigma_t^{1/2} = \frac{1}{n_1}X_t^\top X_t \preceq 1.1\Sigma_t.$$

The proof is finished by taking a union bound over all $t \in [T]$. $\qquad\square$

**Claim A.2** (covariance concentration of target task). *Suppose $n_2 \gg \rho^4(k + \log(1/\delta))$ for $\delta \in (0,1)$. Then for any given matrix $B \in \mathbb{R}^{d \times 2k}$ that is independent of $X_{T+1}$, with probability at least $1 - \frac{\delta}{10}$ over $X_{T+1}$ we have*

$$0.9B^\top \Sigma_{T+1}B \preceq \frac{1}{n_2}B^\top X_{T+1}^\top X_{T+1}B \preceq 1.1B^\top \Sigma_{T+1}B. \tag{23}$$

*Proof.* According to our assumption on $p_{T+1}$, we can write $X_{T+1} = \bar{X}_{T+1}\Sigma_{T+1}^{1/2}$, where $\bar{X}_{T+1} \in \mathbb{R}^{n_2 \times d}$ and the rows of $\bar{X}_{T+1}$ hold i.i.d. samples of $\bar{p}_{T+1}$. We take the SVD of $\Sigma_{T+1}^{1/2}B$: $\Sigma_{T+1}^{1/2}B = UDV^\top$, where $U \in \mathbb{R}^{d \times 2k}$ has orthonormal columns. Now we look at the matrix $\bar{X}_{T+1}U \in \mathbb{R}^{n_2 \times 2k}$. It is easy to see that the rows of $\bar{X}_{T+1}U$ are i.i.d. $2k$-dimensional random vectors with zero mean, identity covariance, and are $\rho^2$-subgaussian. Therefore, applying Lemma A.6, with probability at least $1 - \frac{\delta}{10}$ we have

$$0.9I \preceq \frac{1}{n_2}U^\top \bar{X}_{T+1}^\top \bar{X}_{T+1}U \preceq 1.1I,$$

which implies

$$0.9VDDV^\top \preceq \frac{1}{n_2}VDU^\top \bar{X}_{T+1}^\top \bar{X}_{T+1}UDV^\top \preceq 1.1VDDV^\top.$$

Since $\frac{1}{n_2}VDU^\top \bar{X}_{T+1}^\top \bar{X}_{T+1}UDV^\top = \frac{1}{n_2}B^\top \Sigma_{T+1}^{1/2}\bar{X}_{T+1}^\top \bar{X}_{T+1}\Sigma_{T+1}^{1/2}B = \frac{1}{n_2}B^\top X_{T+1}^\top X_{T+1}B$ and $VDDV^\top = VDU^\top UDV^\top = B^\top \Sigma_{T+1}B$, the above inequality becomes

$$0.9B^\top \Sigma_{T+1}B \preceq \frac{1}{n_2}B^\top X_{T+1}^\top X_{T+1}B \preceq 1.1B^\top \Sigma_{T+1}B. \qquad\square$$

**Claim A.3** (guarantee on source training data). *Under the setting of Theorem 4.1, with probability at least $1 - \frac{\delta}{5}$ we have*

$$\|\mathcal{X}(\hat{B}\hat{W} - B^*W^*)\|_F^2 \lesssim \sigma^2\left(kT + kd\log(\kappa n_1) + \log(1/\delta)\right). \tag{24}$$

*Proof.* We assume that (22) is true, which happens with probability at least $1 - \frac{\delta}{10}$ according to Claim A.1.

Let $\hat{\Theta} = \hat{B}\hat{W}$ and $\Theta^* = B^*W^*$. From the optimality of $\hat{B}$ and $\hat{W}$ for (7) we have $\|Y - \mathcal{X}(\hat{\Theta})\|_F^2 \leq \|Y - \mathcal{X}(\Theta^*)\|_F^2$. Plugging in $Y = \mathcal{X}(\Theta^*) + Z$, this becomes

$$\|\mathcal{X}(\hat{\Theta} - \Theta^*)\|_F^2 \leq 2\langle Z, \mathcal{X}(\hat{\Theta} - \Theta^*)\rangle. \tag{25}$$

Let $\Delta = \hat{\Theta} - \Theta^*$. Since $\mathrm{rank}(\Delta) \le 2k$, we can write $\Delta = VR = [V\boldsymbol{r}_1, \cdots, V\boldsymbol{r}_T]$ where $V \in \mathcal{O}_{d,2k}$ and $R = [\boldsymbol{r}_1, \cdots, \boldsymbol{r}_T] \in \mathbb{R}^{2k \times T}$. Here $\mathcal{O}_{d_1, d_2}$ ($d_1 \ge d_2$) is the set of orthonormal $d_1 \times d_2$ matrices (i.e., the columns are orthonormal). For each $t \in [T]$ we further write $X_t V = U_t Q_t$ where $U_t \in \mathcal{O}_{n_1,2k}$ and $Q_t \in \mathbb{R}^{2k \times 2k}$. Then we have

$$
\begin{aligned}
\langle Z, \mathcal{X}(\Delta) \rangle &= \sum_{t=1}^{T} \boldsymbol{z}_t^\top X_t V \boldsymbol{r}_t \\
&= \sum_{t=1}^{T} \boldsymbol{z}_t^\top U_t Q_t \boldsymbol{r}_t \\
&\le \sum_{t=1}^{T} \left\| U_t^\top \boldsymbol{z}_t \right\| \cdot \left\| Q_t \boldsymbol{r}_t \right\| \\
&\le \sqrt{\sum_{t=1}^{T} \left\| U_t^\top \boldsymbol{z}_t \right\|^2} \cdot \sqrt{\sum_{t=1}^{T} \left\| Q_t \boldsymbol{r}_t \right\|^2} \\
&= \sqrt{\sum_{t=1}^{T} \left\| U_t^\top \boldsymbol{z}_t \right\|^2} \cdot \sqrt{\sum_{t=1}^{T} \left\| U_t Q_t \boldsymbol{r}_t \right\|^2} \\
&= \sqrt{\sum_{t=1}^{T} \left\| U_t^\top \boldsymbol{z}_t \right\|^2} \cdot \sqrt{\sum_{t=1}^{T} \left\| X_t V \boldsymbol{r}_t \right\|^2} \\
&= \sqrt{\sum_{t=1}^{T} \left\| U_t^\top \boldsymbol{z}_t \right\|^2} \cdot \left\| \mathcal{X}(\Delta) \right\|_F .
\end{aligned} \tag{26}
$$

Next we give a high-probability upper bound on $\sum_{t=1}^{T} \left\| U_t^\top \boldsymbol{z}_t \right\|^2$ using the randomness in $Z$. Since $U_t$'s depend on $V$ which depends on $Z$, we will need an $\epsilon$-net argument to cover all possible $V \in \mathcal{O}_{d,2k}$. First, for any fixed $\bar{V} \in \mathcal{O}_{d,2k}$, we let $X_t \bar{V} = \bar{U}_t \bar{Q}_t$ where $\bar{U}_t \in \mathcal{O}_{n,2k}$. The $\bar{U}_t$'s defined in this way are independent of $Z$. Since $Z$ has i.i.d. $\mathcal{N}(0, \sigma^2)$ entries, we know that $\sigma^{-2} \sum_{t=1}^{T} \left\| \bar{U}_t^\top \boldsymbol{z}_t \right\|^2$ is distributed as $\chi^2(2kT)$. Using the standard tail bound for $\chi^2$ random variables, we know that with probability at least $1 - \delta'$ over $Z$,

$$
\sigma^{-2} \sum_{t=1}^{T} \left\| \bar{U}_t^\top \boldsymbol{z}_t \right\|^2 \lesssim kT + \log(1/\delta').
$$

Therefore, using the same argument in (26) we know that with probability at least $1 - \delta'$,

$$
\langle Z, \mathcal{X}(\bar{V}R) \rangle \lesssim \sigma \sqrt{kT + \log(1/\delta')} \left\| \mathcal{X}(\bar{V}R) \right\|_F .
$$

Now, from Lemma A.5 we know that there exists an $\epsilon$-net $\mathcal{N}$ of $\mathcal{O}_{d,2k}$ in Frobenius norm such that $\mathcal{N} \subset \mathcal{O}_{d,2k}$ and $|\mathcal{N}| \le (\frac{6\sqrt{2k}}{\epsilon})^{2kd}$. Applying a union bound over $\mathcal{N}$, we know that with probability at least $1 - \delta'|\mathcal{N}|$,

$$
\langle Z, \mathcal{X}(\bar{V}R) \rangle \lesssim \sigma \sqrt{kT + \log(1/\delta')} \left\| \mathcal{X}(\bar{V}R) \right\|_F, \quad \forall \bar{V} \in \mathcal{N}. \tag{27}
$$

Choosing $\delta' = \frac{\delta}{20(\frac{6\sqrt{2k}}{\epsilon})^{2kd}}$, we know that (27) holds with probability at least $1 - \frac{\delta}{20}$.

We will use (22), (25) and (27) to complete the proof of the claim. This is done in the following steps:

1. Upper bounding $\|Z\|_F$.

   Since $\sigma^{-2} \|Z\|_F^2 \sim \chi^2(n_1 T)$, we know that with probability at least $1 - \frac{\delta}{20}$,

$$
\|Z\|_F^2 \lesssim \sigma^2 (n_1 T + \log(1/\delta)). \tag{28}
$$

2. Upper bounding $\|\Delta\|_F$.

From (25) we have $\|\mathcal{X}(\Delta)\|_F^2 \leq 2\|Z\|_F \|\mathcal{X}(\Delta)\|_F$, which implies $\|\mathcal{X}(\Delta)\|_F \leq 2\|Z\|_F \lesssim \sigma\sqrt{n_1 T + \log(1/\delta)}$. On the other hand, letting the $t$-th column of $\Delta$ be $\boldsymbol{\delta}_t$, we have

$$
\begin{aligned}
\|\mathcal{X}(\Delta)\|_F^2 &= \sum_{t=1}^T \|X_t \boldsymbol{\delta}_t\|^2 \\
&= \sum_{t=1}^T \boldsymbol{\delta}_t^\top X_t^\top X_t \boldsymbol{\delta}_t \\
&\geq 0.9 n_1 \sum_{t=1}^T \boldsymbol{\delta}_t^\top \Sigma_t \boldsymbol{\delta}_t && \text{(using (22))} \\
&\geq 0.9 n_1 \sum_{t=1}^T \lambda_{\min}(\Sigma_t) \|\boldsymbol{\delta}_t\|^2 \\
&\geq 0.9 n_1 \underline{\lambda} \|\Delta\|_F^2 ,
\end{aligned}
$$

where $\underline{\lambda} = \min_{t \in [T]} \lambda_{\min}(\Sigma_t)$. Hence we obtain

$$
\|\Delta\|_F^2 \lesssim \frac{\|\mathcal{X}(\Delta)\|_F^2}{n_1 \underline{\lambda}} \lesssim \frac{\sigma^2(n_1 T + \log(1/\delta))}{n_1 \underline{\lambda}}.
$$

3. Applying the $\epsilon$-net $\mathcal{N}$.

Let $\bar{V} \in \mathcal{N}$ such that $\|V - \bar{V}\|_F \leq \epsilon$. Then we have

$$
\begin{aligned}
&\left\|\mathcal{X}(VR - \bar{V}R)\right\|_F^2 \\
&= \sum_{t=1}^T \left\|X_t(V - \bar{V})\boldsymbol{r}_t\right\|^2 \\
&\leq \sum_{t=1}^T \|X_t\|^2 \|V - \bar{V}\|^2 \|\boldsymbol{r}_t\|^2 \\
&\leq \sum_{t=1}^T 1.1 n_1 \lambda_{\max}(\Sigma_t) \epsilon^2 \|\boldsymbol{r}_t\|^2 && \text{(using (22))} \\
&\leq 1.1 n_1 \bar{\lambda} \epsilon^2 \|R\|_F^2 && (\bar{\lambda} = \max_{t \in [T]} \lambda_{\max}(\Sigma_t)) \\
&= 1.1 n_1 \bar{\lambda} \epsilon^2 \|\Delta\|_F^2 && (\|\Delta\|_F = \|VR\|_F = \|R\|_F) \\
&\lesssim n_1 \bar{\lambda} \epsilon^2 \cdot \frac{\sigma^2(n_1 T + \log(1/\delta))}{n_1 \underline{\lambda}} \\
&= \kappa \epsilon^2 \sigma^2 (n_1 T + \log(1/\delta)). && (29)
\end{aligned}
$$

4. Finishing the proof.

We have the following chain of inequalities:

$$
\begin{aligned}
&\frac{1}{2} \|\mathcal{X}(\Delta)\|_F^2 \\
&\leq \langle Z, \mathcal{X}(\Delta) \rangle && \text{(using (25))} \\
&= \langle Z, \mathcal{X}(\bar{V}R) \rangle + \langle Z, \mathcal{X}(VR - \bar{V}R) \rangle \\
&\lesssim \sigma\sqrt{kT + \log(1/\delta')} \left\|\mathcal{X}(\bar{V}R)\right\|_F + \|Z\|_F \left\|\mathcal{X}(VR - \bar{V}R)\right\|_F && \text{(using (27))} \\
&\leq \sigma\sqrt{kT + \log(1/\delta')} \left(\|\mathcal{X}(VR)\|_F + \left\|\mathcal{X}(VR - \bar{V}R)\right\|_F\right) \\
&\quad + \sigma\sqrt{n_1 T + \log(1/\delta)} \left\|\mathcal{X}(VR - \bar{V}R)\right\|_F && \text{(using (28))}
\end{aligned}
$$

$$\lesssim \sigma \sqrt{kT + \log(1/\delta')} \left\| \mathcal{X}(VR) \right\|_F + \sigma \sqrt{n_1 T + \log(1/\delta')} \left\| \mathcal{X}(VR - \bar{V}R) \right\|_F$$
$$\text{(using } k < n_1 \text{ and } \delta' < \delta )$$

$$\lesssim \sigma \sqrt{kT + \log(1/\delta')} \left\| \mathcal{X}(\Delta) \right\|_F + \sigma \sqrt{n_1 T + \log(1/\delta')} \cdot \sqrt{\kappa \epsilon^2 \sigma^2 (n_1 T + \log(1/\delta))}$$
$$\text{(using (29))}$$

$$\leq \sigma \sqrt{kT + \log(1/\delta')} \left\| \mathcal{X}(\Delta) \right\|_F + \epsilon \sigma^2 \sqrt{\kappa}(n_1 T + \log(1/\delta')).$$

Finally, we let $\epsilon = \frac{k}{\sqrt{\kappa} n_1}$, and recall $\delta' = \frac{\delta}{20 (\frac{6\sqrt{2k}}{\epsilon})^{2kd}}$. Then the above inequality implies

$$\left\| \mathcal{X}(\Delta) \right\|_F$$

$$\lesssim \max \left\{ \sigma \sqrt{kT + \log(1/\delta')}, \sqrt{\epsilon \sigma^2 \sqrt{\kappa}(n_1 T + \log(1/\delta'))} \right\}$$

$$= \max \left\{ \sigma \sqrt{kT + \log(1/\delta')}, \sigma \sqrt{\frac{k}{n_1}(n_1 T + \log(1/\delta'))} \right\}$$

$$\leq \max \left\{ \sigma \sqrt{kT + \log(1/\delta')}, \sigma \sqrt{kT + \log(1/\delta')} \right\} \qquad \text{(using } k < n_1)$$

$$= \sigma \sqrt{kT + \log(1/\delta')}$$

$$\lesssim \sigma \sqrt{kT + kd \log \frac{k}{\epsilon} + \log \frac{1}{\delta}}$$

$$\leq \sigma \sqrt{kT + kd \log(\kappa n_1) + \log \frac{1}{\delta}}.$$

The high-probability events we have used in the proof are (22), (27) and (28). By a union bound, the failure probability is at most $\frac{\delta}{10} + \frac{\delta}{20} + \frac{\delta}{20} = \frac{\delta}{5}$. Therefore the proof is completed. $\qquad \square$

**Claim A.4** (Guarantee on target training data). *Under the setting of Theorem 4.1, with probability at least $1 - \frac{2\delta}{5}$, we have*

$$\frac{1}{n_2} \left\| P^{\perp}_{X_{T+1}\hat{B}} X_{T+1} B^* \right\|_F^2 \lesssim \frac{\sigma^2 \left( kT + kd \log(\kappa n_1) + \log \frac{1}{\delta} \right)}{cn_1 \cdot \sigma_k^2(W^*)}.$$

*Proof.* We suppose that the high-probability events in Claims A.1, A.2 and A.3 happen, which holds with probability at least $1 - \frac{2\delta}{5}$. Here we instantiate Claim A.2 using $B = [\hat{B}, B^*] \in \mathbb{R}^{d \times 2k}$.

From the optimality of $\hat{B}$ and $\hat{W}$ in (6) we know $X_t \hat{B} \hat{w}_t = P_{X_t \hat{B}} y_t = P_{X_t \hat{B}}(X_t B^* w_t^* + z_t)$ for each $t \in [T]$. Then we have

$$\sigma^2 \left( kT + kd \log(\kappa n_1) + \log(1/\delta) \right)$$

$$\gtrsim \left\| \mathcal{X}(\hat{B}\hat{W} - B^* W^*) \right\|_F^2 \qquad \text{(from (24))}$$

$$= \sum_{t=1}^{T} \left\| X_t \hat{B} \hat{w}_t - X_t B^* \hat{w}_t^* \right\|^2$$

$$= \sum_{t=1}^{T} \left\| P_{X_t \hat{B}}(X_t B^* w_t^* + z_t) - X_t B^* \hat{w}_t^* \right\|^2$$

$$= \sum_{t=1}^{T} \left\| -P^{\perp}_{X_t \hat{B}} X_t B^* w_t^* + P_{X_t \hat{B}} z_t \right\|^2$$

$$= \sum_{t=1}^{T} \left( \left\| -P^{\perp}_{X_t \hat{B}} X_t B^* w_t^* \right\|^2 + \left\| P_{X_t \hat{B}} z_t \right\|^2 \right) \qquad \text{(the cross term is 0)}$$

$$\geq \sum_{t=1}^{T} \left\| P^{\perp}_{X_t \hat{B}} X_t B^* w_t^* \right\|^2$$

$$\geq 0.9n_1 \sum_{t=1}^{T} \left\| P_{\Sigma_t^{1/2}\hat{B}}^{\perp} \Sigma_t^{1/2} B^* \boldsymbol{w}_t^* \right\|^2 \qquad \text{(using (22) and Lemma A.7)}$$

$$\geq 0.9cn_1 \sum_{t=1}^{T} \left\| P_{\Sigma_{T+1}^{1/2}\hat{B}}^{\perp} \Sigma_{T+1}^{1/2} B^* \boldsymbol{w}_t^* \right\|^2 \qquad \text{(using Assumption 4.2 and Lemma A.7)}$$

$$= 0.9cn_1 \left\| P_{\Sigma_{T+1}^{1/2}\hat{B}}^{\perp} \Sigma_{T+1}^{1/2} B^* W^* \right\|_F^2$$

$$\geq 0.9cn_1 \left\| P_{\Sigma_{T+1}^{1/2}\hat{B}}^{\perp} \Sigma_{T+1}^{1/2} B^* \right\|_F^2 \cdot \sigma_k^2(W^*).$$

Next, we write $\hat{B} = [\hat{B}, B^*] \begin{bmatrix} I \\ 0 \end{bmatrix} =: BA$ and $B^* = [\hat{B}, B^*] \begin{bmatrix} 0 \\ I \end{bmatrix} =: BC$. Recall that we have $\frac{1}{n_2} B^\top X_{T+1}^\top X_{T+1} B \preceq 1.1 B^\top \Sigma_{T+1} B$ from Claim A.2. Then using Lemma A.7 we can obtain

$$1.1 \left\| P_{\Sigma_{T+1}^{1/2}BA}^{\perp} \Sigma_{T+1}^{1/2} BC \right\|_F^2 \geq \frac{1}{n_2} \left\| P_{X_{T+1}BA}^{\perp} X_{T+1} BC \right\|_F^2,$$

i.e.,

$$1.1 \left\| P_{\Sigma_{T+1}^{1/2}\hat{B}}^{\perp} \Sigma_{T+1}^{1/2} B^* \right\|_F^2 \geq \frac{1}{n_2} \left\| P_{X_{T+1}\hat{B}}^{\perp} X_{T+1} B^* \right\|_F^2.$$

Therefore we get

$$\sigma^2 \left( kT + kd \log(\kappa n_1) + \log(1/\delta) \right) \gtrsim \frac{0.9cn_1}{1.1n_2} \left\| P_{X_{T+1}\hat{B}}^{\perp} X_{T+1} B^* \right\|_F^2 \cdot \sigma_k^2(W^*),$$

completing the proof. $\qquad \square$

*Proof of Theorem 4.1.* We will use all the high-probability events in Claims A.1, A.2, A.3 and A.4. Here we instantiate Claim A.2 using $B = [\hat{B}, B^*] \in \mathbb{R}^{d \times 2k}$. The success probability is at least $1 - \frac{4\delta}{5}$.

For the target task, the excess risk of our learned linear predictor $\boldsymbol{x} \mapsto (\hat{B}\hat{\boldsymbol{w}}_{T+1})^\top \boldsymbol{x}$ is

$$\mathrm{ER}(\hat{B}, \hat{\boldsymbol{w}}_{T+1}) = \frac{1}{2} \mathbb{E}_{\boldsymbol{x} \sim p_{T+1}} \left[ \left( \boldsymbol{x}^\top (\hat{B}\hat{\boldsymbol{w}}_{T+1} - B^* \boldsymbol{w}_{T+1}^*) \right)^2 \right]$$

$$= \frac{1}{2} (\hat{B}\hat{\boldsymbol{w}}_{T+1} - B^* \boldsymbol{w}_{T+1}^*)^\top \Sigma_{T+1} (\hat{B}\hat{\boldsymbol{w}}_{T+1} - B^* \boldsymbol{w}_{T+1}^*).$$

Applying Claim A.2 with $B = [\hat{B}, B^*]$, we have

$$0.9 B^\top \Sigma_{T+1} B \preceq \frac{1}{n_2} B^\top X_{T+1}^\top X_{T+1} B,$$

which implies $0.9 \boldsymbol{v}^\top B^\top \Sigma_{T+1} B \boldsymbol{v} \leq \frac{1}{n_2} \boldsymbol{v} B^\top X_{T+1}^\top X_{T+1} B \boldsymbol{v}$ for $\boldsymbol{v} = \begin{bmatrix} \hat{\boldsymbol{w}}_{T+1} \\ \boldsymbol{w}_{T+1}^* \end{bmatrix}$. This becomes

$$(\hat{B}\hat{\boldsymbol{w}}_{T+1} - B^* \boldsymbol{w}_{T+1}^*)^\top \Sigma_{T+1} (\hat{B}\hat{\boldsymbol{w}}_{T+1} - B^* \boldsymbol{w}_{T+1}^*)$$

$$\leq \frac{1}{0.9n_2} (\hat{B}\hat{\boldsymbol{w}}_{T+1} - B^* \boldsymbol{w}_{T+1}^*)^\top X_{T+1}^\top X_{T+1} (\hat{B}\hat{\boldsymbol{w}}_{T+1} - B^* \boldsymbol{w}_{T+1}^*).$$

Therefore we have

$$\mathrm{ER}(\hat{B}, \hat{\boldsymbol{w}}_{T+1}) \leq \frac{1}{1.8n_2} (\hat{B}\hat{\boldsymbol{w}}_{T+1} - B^* \boldsymbol{w}_{T+1}^*)^\top X_{T+1}^\top X_{T+1} (\hat{B}\hat{\boldsymbol{w}}_{T+1} - B^* \boldsymbol{w}_{T+1}^*)$$

$$= \frac{1}{1.8n_2} \left\| X_{T+1}(\hat{B}\hat{\boldsymbol{w}}_{T+1} - B^* \boldsymbol{w}_{T+1}^*) \right\|^2.$$

From the optimality of $\hat{w}_{T+1}$ in (8) we know $X_{T+1}\hat{B}\hat{w}_{T+1} = P_{X_{T+1}\hat{B}}\boldsymbol{y}_{T+1} = P_{X_{T+1}\hat{B}}(X_{T+1}B^*\boldsymbol{w}_{T+1}^* + \boldsymbol{z}_{T+1})$. It follows that

$$
\begin{aligned}
\mathrm{ER}(\hat{B}, \hat{w}_{T+1}) &\lesssim \frac{1}{n_2}\left\| P_{X_{T+1}\hat{B}}(X_{T+1}B^*\boldsymbol{w}_{T+1}^* + \boldsymbol{z}_{T+1}) - X_{T+1}B^*\boldsymbol{w}_{T+1}^* \right\|_F^2 \\
&= \frac{1}{n_2}\left\| -P_{X_{T+1}\hat{B}}^{\perp}X_{T+1}B^*\boldsymbol{w}_{T+1}^* + P_{X_{T+1}\hat{B}}\boldsymbol{z}_{T+1} \right\|_F^2 \\
&= \frac{1}{n_2}\left\| P_{X_{T+1}\hat{B}}^{\perp}X_{T+1}B^*\boldsymbol{w}_{T+1}^* \right\|_F^2 + \frac{1}{n_2}\left\| P_{X_{T+1}\hat{B}}\boldsymbol{z}_{T+1} \right\|_F^2 .
\end{aligned}
$$

Recall that $\boldsymbol{w}_{T+1}^* \sim \nu$ and $\left\|\mathbb{E}_{\boldsymbol{w}\sim\nu}[\boldsymbol{w}\boldsymbol{w}^\top]\right\| \leq O(\frac{1}{k})$. Taking expectation over $\boldsymbol{w}_{T+1}^* \sim \nu$ and denoting $\Sigma = \mathbb{E}_{\boldsymbol{w}\sim\nu}[\boldsymbol{w}\boldsymbol{w}^\top]$, we obtain

$$
\begin{aligned}
&\mathbb{E}_{\boldsymbol{w}_{T+1}^*\sim\nu}[\mathrm{ER}(\hat{B}, \hat{w}_{T+1})] \\
&\lesssim \frac{1}{n_2}\mathbb{E}_{\boldsymbol{w}_{T+1}^*\sim\nu}\left[\left\| P_{X_{T+1}\hat{B}}^{\perp}X_{T+1}B^*\boldsymbol{w}_{T+1}^* \right\|_F^2\right] + \frac{1}{n_2}\left\| P_{X_{T+1}\hat{B}}\boldsymbol{z}_{T+1} \right\|_F^2 \\
&= \frac{1}{n_2}\mathbb{E}_{\boldsymbol{w}_{T+1}^*\sim\nu}\left[\mathrm{Tr}\left[ P_{X_{T+1}\hat{B}}^{\perp}X_{T+1}B^*\boldsymbol{w}_{T+1}^*\boldsymbol{w}_{T+1}^{*\top}\left(P_{X_{T+1}\hat{B}}^{\perp}X_{T+1}B^*\right)^\top\right]\right] + \frac{1}{n_2}\left\| P_{X_{T+1}\hat{B}}\boldsymbol{z}_{T+1} \right\|_F^2 \\
&= \frac{1}{n_2}\mathrm{Tr}\left[ P_{X_{T+1}\hat{B}}^{\perp}X_{T+1}B^*\Sigma\left(P_{X_{T+1}\hat{B}}^{\perp}X_{T+1}B^*\right)^\top\right] + \frac{1}{n_2}\left\| P_{X_{T+1}\hat{B}}\boldsymbol{z}_{T+1} \right\|_F^2 \\
&= \frac{1}{n_2}\left\| P_{X_{T+1}\hat{B}}^{\perp}X_{T+1}B^*\Sigma^{1/2} \right\|_F^2 + \frac{1}{n_2}\left\| P_{X_{T+1}\hat{B}}\boldsymbol{z}_{T+1} \right\|_F^2 \\
&\leq \frac{1}{n_2}\left\| P_{X_{T+1}\hat{B}}^{\perp}X_{T+1}B^* \right\|_F^2 \left\| \Sigma^{1/2} \right\|^2 + \frac{1}{n_2}\left\| P_{X_{T+1}\hat{B}}\boldsymbol{z}_{T+1} \right\|_F^2 \\
&\lesssim \frac{1}{n_2 k}\left\| P_{X_{T+1}\hat{B}}^{\perp}X_{T+1}B^* \right\|_F^2 + \frac{1}{n_2}\left\| P_{X_{T+1}\hat{B}}\boldsymbol{z}_{T+1} \right\|_F^2 && \text{(using } \|\Sigma\| \lesssim \tfrac{1}{k}) \\
&\lesssim \frac{1}{k}\cdot\frac{\sigma^2\left(kT + kd\log(\kappa n_1) + \log(1/\delta)\right)}{cn_1 \cdot \sigma_k^2(W^*)} + \frac{1}{n_2}\left\| P_{X_{T+1}\hat{B}}\boldsymbol{z}_{T+1} \right\|_F^2 && \text{(using Claim A.4)} \\
&\lesssim \frac{\sigma^2\left(kT + kd\log(\kappa n_1) + \log(1/\delta)\right)}{cn_1 T} + \frac{1}{n_2}\left\| P_{X_{T+1}\hat{B}}\boldsymbol{z}_{T+1} \right\|_F^2 . && \text{(using } \sigma_k^2(W^*) \gtrsim \tfrac{T}{k})
\end{aligned}
$$

For the second term above, notice that $\frac{1}{\sigma^2}\left\| P_{X_{T+1}\hat{B}}\boldsymbol{z}_{T+1} \right\|_F^2 \sim \chi^2(k)$, and thus with probability at least $1 - \frac{\delta}{5}$ we have $\frac{1}{\sigma^2}\left\| P_{X_{T+1}\hat{B}}\boldsymbol{z}_{T+1} \right\|_F^2 \lesssim k + \log\frac{1}{\delta}$. Therefore we obtain the final bound

$$
\begin{aligned}
\mathbb{E}_{\boldsymbol{w}_{T+1}^*\sim\nu}[\mathrm{ER}(\hat{B}, \hat{w}_{T+1})] &\lesssim \frac{\sigma^2\left(kT + kd\log(\kappa n_1) + \log(1/\delta)\right)}{cn_1 T} + \frac{\sigma^2(k + \log\frac{1}{\delta})}{n_2} \\
&= \sigma^2\left(\frac{kd\log(\kappa n_1)}{cn_1 T} + \frac{k}{cn_1} + \frac{\log\frac{1}{\delta}}{cn_1 T} + \frac{k + \log\frac{1}{\delta}}{n_2}\right) \\
&\lesssim \sigma^2\left(\frac{kd\log(\kappa n_1)}{cn_1 T} + \frac{k + \log\frac{1}{\delta}}{n_2}\right),
\end{aligned}
$$

where the last inequality is due to $cn_1 \geq n_2$. $\qquad\square$

## A.1 TECHNICAL LEMMAS

**Lemma A.5.** *Let $\mathcal{O}_{d_1,d_2} = \{V \in \mathbb{R}^{d_1 \times d_2} \mid V^\top V = I\}$ ($d_1 \geq d_2$), and $\epsilon \in (0,1)$. Then there exists a subset $\mathcal{N} \subset \mathcal{O}_{d_1,d_2}$ that is an $\epsilon$-net of $\mathcal{O}_{d_1,d_2}$ in Frobenius norm such that $|\mathcal{N}| \leq (\frac{6\sqrt{d_2}}{\epsilon})^{d_1 d_2}$, i.e., for any $V \in \mathcal{O}_{d_1,d_2}$, there exists $V' \in \mathcal{N}$ such that $\|V - V'\|_F \leq \epsilon$.*

*Proof.* For any $V \in \mathcal{O}_{d_1,d_2}$, each column of $V$ has unit $\ell_2$ norm. It is well known that there exists an $\frac{\epsilon}{2\sqrt{d_2}}$-net (in $\ell_2$ norm) of the unit sphere in $\mathbb{R}^{d_1}$ with size $(\frac{6\sqrt{d_2}}{\epsilon})^{d_1}$. Using this net to cover

all the columns, we obtain a set $\mathcal{N}' \subset \mathbb{R}^{d_1 \times d_2}$ that is an $\frac{\epsilon}{2}$-net of $\mathcal{O}_{d_1,d_2}$ in Frobenius norm and $|\mathcal{N}'| \leq (\frac{6\sqrt{d_2}}{\epsilon})^{d_1 d_2}$.

Finally, we need to transform $\mathcal{N}'$ into an $\epsilon$-net $\mathcal{N}$ that is a subset of $\mathcal{O}_{d_1,d_2}$. This can be done by projecting each point in $\mathcal{N}'$ onto $\mathcal{O}_{d_1,d_2}$. Namely, for each $\bar{V} \in \mathcal{N}'$, let $\mathcal{P}(\bar{V})$ be its closest point in $\mathcal{O}_{d_1,d_2}$ (in Frobenium norm); then define $\mathcal{N} = \{\mathcal{P}(\bar{V}) \mid \bar{V} \in \mathcal{N}'\}$. Then we have $|\mathcal{N}| \leq |\mathcal{N}'| \leq (\frac{6\sqrt{d_2}}{\epsilon})^{d_1 d_2}$ and $\mathcal{N}$ is an $\epsilon$-net of $\mathcal{O}_{d_1,d_2}$, because for any $V \in \mathcal{O}_{d_1,d_2}$, there exists $\bar{V} \in \mathcal{N}'$ such that $\|V - \bar{V}\|_F \leq \frac{\epsilon}{2}$, which implies $\mathcal{P}(\bar{V}) \in \mathcal{N}$ and $\|V - \mathcal{P}(\bar{V})\|_F \leq \|V - \bar{V}\|_F + \|\bar{V} - \mathcal{P}(\bar{V})\|_F \leq \|V - \bar{V}\|_F + \|\bar{V} - V\|_F = 2\|V - \bar{V}\|_F \leq \epsilon$. $\qquad \square$

**Lemma A.6.** *Let $a_1, \ldots, a_n$ be i.i.d. $d$-dimensional random vectors such that $\mathbb{E}[a_i] = 0$, $\mathbb{E}[a_i a_i^\top] = I$, and $a_i$ is $\rho^2$-subgaussian. For $\delta \in (0,1)$, suppose $n \gg \rho^4(d + \log(1/\delta))$. Then with probability at least $1 - \delta$ we have*

$$0.9I \preceq \frac{1}{n}\sum_{i=1}^{n} a_i a_i^\top \preceq 1.1I.$$

*Proof.* Let $A = \frac{1}{n}\sum_{i=1}^{n} a_i a_i^\top - I$. Then it suffices to show $\|A\| \leq 0.1$ with probability at least $1 - \delta$.

We use a standard $\epsilon$-net argument for the unit sphere $\mathcal{S}^{d-1} = \{v \in \mathbb{R}^d : \|v\| = 1\}$. First, consider any fixed $v \in \mathcal{S}^{d-1}$. We have $v^\top A v = \frac{1}{n}\sum_{i=1}^{n}[(v^\top a_i)^2 - 1]$. From our assumptions on $a_i$ we know that $v^\top a_i$ has mean 0 and variance 1 and is $\rho^2$-subgaussian. (Note that we must have $\rho \geq 1$.) Therefore $(v^\top a_i)^2 - 1$ is zero-mean and $16\rho^2$-sub-exponential. By Bernstein inequality for sub-exponential random variables, we have for any $\epsilon > 0$,

$$\Pr\left[|v^\top A v| > \epsilon\right] \leq 2\exp\left(-\frac{n}{2}\min\left\{\frac{\epsilon^2}{(16\rho^2)^2}, \frac{\epsilon}{16\rho^2}\right\}\right).$$

Next, take a $\frac{1}{5}$-net $\mathcal{N} \subset \mathcal{S}^{d-1}$ of $\mathcal{S}^{d-1}$ with size $|\mathcal{N}| \leq e^{O(d)}$. By a union bound over all $v \in \mathcal{N}$, we have

$$\Pr\left[\max_{v \in \mathcal{N}}|v^\top A v| > \epsilon\right] \leq 2|\mathcal{N}|\exp\left(-\frac{n}{2}\min\left\{\frac{\epsilon^2}{(16\rho^2)^2}, \frac{\epsilon}{16\rho^2}\right\}\right)$$

$$\leq \exp\left(O(d) - \frac{n}{2}\min\left\{\frac{\epsilon^2}{(16\rho^2)^2}, \frac{\epsilon}{16\rho^2}\right\}\right).$$

Plugging in $\epsilon = \frac{1}{20}$ and noticing $\rho > 1$, the above inequality becomes

$$\Pr\left[\max_{v \in \mathcal{N}}|v^\top A v| > \frac{1}{20}\right] \leq \exp\left(O(d) - \frac{n}{2}\cdot\frac{(1/20)^2}{(16\rho^2)^2}\right) \leq \delta,$$

where the last inequality is due to $n \gg \rho^4(d + \log(1/\delta))$.

Therefore, with probability at least $1 - \delta$ we have $\max_{v \in \mathcal{N}}|v^\top A v| \leq \frac{1}{20}$. Suppose this indeed happens. Next, for any $u \in \mathcal{S}^{d-1}$, there exists $u' \in \mathcal{N}$ such that $\|u - u'\| \leq \frac{1}{5}$. Then we have

$$\|u^\top A u\| \leq \|(u')^\top A u'\| + 2\|(u - u')^\top A u'\| + \|(u - u')^\top A(u - u')\|$$

$$\leq \frac{1}{20} + 2\|u - u'\|\cdot\|A\|\cdot\|u'\| + \|u - u'\|^2\cdot\|A\|$$

$$\leq \frac{1}{20} + 2\cdot\frac{1}{5}\cdot\|A\|\cdot 1 + \left(\frac{1}{5}\right)^2\cdot\|A\|$$

$$\leq \frac{1}{20} + \frac{1}{2}\|A\|.$$

Taking a supreme over $u \in \mathcal{S}^{d-1}$, we obtain $\|A\| \leq \frac{1}{20} + \frac{1}{2}\|A\|$, i.e., $\|A\| \leq \frac{1}{10}$. $\qquad \square$

**Lemma A.7.** *If two matrices $A_1$ and $A_2$ (with the same number of columns) satisfy $A_1^\top A_1 \succeq A_2^\top A_2$, then for any matrix $B$ (of compatible dimensions), we have*

$$A_1^\top P_{A_1 B}^\perp A_1 \succeq A_2^\top P_{A_2 B}^\perp A_2.$$

*As a consequence, for any matrices $B$ and $B'$ (of compatible dimensions), we have*

$$\left\|P_{A_1 B}^{\perp} A_1 B'\right\|_F^2 \geq \left\|P_{A_2 B}^{\perp} A_2 B'\right\|_F^2.$$

*Proof.* For the first part of the lemma, it suffices to show the following for any vector $\boldsymbol{v}$:

$$\boldsymbol{v}^{\top} A_1^{\top} P_{A_1 B}^{\perp} A_1 \boldsymbol{v} \geq \boldsymbol{v}^{\top} A_2^{\top} P_{A_2 B}^{\perp} A_2 \boldsymbol{v},$$

which is equivalent to

$$\min_{\boldsymbol{w}} \|A_1 B \boldsymbol{w} - A_1 \boldsymbol{v}\|_2^2 \geq \min_{\boldsymbol{w}} \|A_2 B \boldsymbol{w} - A_2 \boldsymbol{v}\|_2^2.$$

Let $\boldsymbol{w}^* \in \arg\min_{\boldsymbol{w}} \|A_1 B \boldsymbol{w} - A_1 \boldsymbol{v}\|_2^2$. Then we have

$$
\begin{aligned}
\min_{\boldsymbol{w}} \|A_1 B \boldsymbol{w} - A_1 \boldsymbol{v}\|_2^2 &= \|A_1 B \boldsymbol{w}^* - A_1 \boldsymbol{v}\|_2^2 \\
&= (B \boldsymbol{w}^* - \boldsymbol{v})^{\top} A_1^{\top} A_1 (B \boldsymbol{w}^* - \boldsymbol{v}) \\
&\geq (B \boldsymbol{w}^* - \boldsymbol{v})^{\top} A_2^{\top} A_2 (B \boldsymbol{w}^* - \boldsymbol{v}) \\
&= \|A_2 B \boldsymbol{w}^* - A_2 \boldsymbol{v}\|_2^2 \\
&\geq \min_{\boldsymbol{w}} \|A_2 B \boldsymbol{w} - A_2 \boldsymbol{v}\|_2^2,
\end{aligned}
$$

finishing the proof of the first part.

For the second part, from $A_1^{\top} P_{A_1 B}^{\perp} A_1 \succeq A_2^{\top} P_{A_2 B}^{\perp} A_2$ we know

$$(B')^{\top} A_1^{\top} P_{A_1 B}^{\perp} A_1 B' \succeq (B')^{\top} A_2^{\top} P_{A_2 B}^{\perp} A_2 B'.$$

Taking trace on both sides, we obtain

$$\left\|P_{A_1 B}^{\perp} A_1 B'\right\|_F^2 \geq \left\|P_{A_2 B}^{\perp} A_2 B'\right\|_F^2,$$

which finishes the proof. □

## B  PROOF OF THEOREM 5.1

Here we first prove an important intermediate result on the in-sample risk, which explains how the Gaussian width of $\mathcal{F}_{\mathcal{X}}(\Phi)$ arises.

**Claim B.1** (analogue of Claim A.3). *Let $\hat{\phi}$ and $\hat{\boldsymbol{w}}_1, \ldots, \hat{\boldsymbol{w}}_T$ be the optimal solution to (2). Then with probability at least $1 - \delta$ we have*

$$\sum_{t=1}^{T} \left\|\hat{\phi}(X_t)\hat{\boldsymbol{w}}_t - \phi^*(X_t)\boldsymbol{w}_t^*\right\|^2 \lesssim \sigma^2 \left(\mathcal{G}(\mathcal{F}_{\mathcal{X}}(\Phi))^2 + \log \frac{1}{\delta}\right).$$

*Proof.* By the optimality of $\hat{\phi}$ and $\hat{\boldsymbol{w}}_1, \ldots, \hat{\boldsymbol{w}}_T$ for (2), we know

$$\sum_{t=1}^{T} \left\|\boldsymbol{y}_t - \hat{\phi}(X_t)\hat{\boldsymbol{w}}_t\right\|^2 \leq \sum_{t=1}^{T} \|\boldsymbol{y}_t - \phi^*(X_t)\boldsymbol{w}_t^*\|^2.$$

Plugging in $\boldsymbol{y}_t = \phi^*(X_t)\boldsymbol{w}_t^* + \boldsymbol{z}_t$ ($\boldsymbol{z}_t \sim \mathcal{N}(0, I)$ is independent of $X_t$), we get

$$\sum_{t=1}^{T} \left\|\phi^*(X_t)\boldsymbol{w}_t^* + \boldsymbol{z}_t - \hat{\phi}(X_t)\hat{\boldsymbol{w}}_t\right\|^2 \leq \sum_{t=1}^{T} \|\boldsymbol{z}_t\|^2,$$

which gives

$$\sum_{t=1}^{T} \left\|\hat{\phi}(X_t)\hat{\boldsymbol{w}}_t - \phi^*(X_t)\boldsymbol{w}_t^*\right\|^2 \leq 2 \sum_{t=1}^{t} \langle \boldsymbol{z}_t, \hat{\phi}(X_t)\hat{\boldsymbol{w}}_t - \phi^*(X_t)\boldsymbol{w}_t^* \rangle.$$

Denote $Z = [\boldsymbol{z}_1, \cdots, \boldsymbol{z}_T] \in \mathbb{R}^{n_1 \times T}$ and $A = [\boldsymbol{a}_1, \cdots, \boldsymbol{a}_T] \in \mathbb{R}^{n_1 \times T}$ where $\boldsymbol{a}_t = \hat{\phi}(X_t)\hat{\boldsymbol{w}}_t - \phi^*(X_t)\boldsymbol{w}_t^*$. Then the above inequality reads $\|A\|_F^2 \leq 2\langle Z, A \rangle$. Notice that $\frac{A}{\|A\|_F} \in \mathcal{F}_{\mathcal{X}}(\Phi)$ (c.f. (10)). It follows that

$$\|A\|_F \leq 2 \left\langle Z, \frac{A}{\|A\|_F} \right\rangle \leq 2 \sup_{\bar{A} \in \mathcal{F}_{\mathcal{X}}(\Phi)} \langle Z, \bar{A} \rangle. \tag{30}$$

By definition, we have $\mathbb{E}_Z \left[ \sup_{\bar{A} \in \mathcal{F}_{\mathcal{X}}(\Phi)} \langle \sigma^{-1} Z, \bar{A} \rangle \right] = \mathcal{G}(\mathcal{F}_{\mathcal{X}}(\Phi))$. Furthermore, since the function $Z \mapsto \sup_{\bar{A} \in \mathcal{F}_{\mathcal{X}}(\Phi)} \langle Z, \bar{A} \rangle$ is 1-Lipschitz in Frobenius norm, by the standard Gaussian concentration inequality, we have with probability at least $1 - \delta$,

$$\sup_{\bar{A} \in \mathcal{F}_{\mathcal{X}}(\Phi)} \langle \sigma^{-1} Z, \bar{A} \rangle \leq \mathbb{E} \left[ \sup_{\bar{A} \in \mathcal{F}_{\mathcal{X}}(\Phi)} \langle \sigma^{-1} Z, \bar{A} \rangle \right] + \sqrt{\log \frac{1}{\delta}} = \mathcal{G}(\mathcal{F}_{\mathcal{X}}(\Phi)) + \sqrt{\log \frac{1}{\delta}}.$$

Then the proof is completed using (30). $\qquad \square$

The proof is conditioned on several high-probability events, each happening with probability at least $1 - \Omega(\delta)$. By a union bound at the end, the final success probability is also at least $1 - \Omega(\delta)$. We can always rescale $\delta$ by a constant factor such that the final probability is at least $1 - \delta$. Therefore, we will not carefully track the constants before $\delta$ in the proof. All the $\delta$'s should be understood as $\Omega(\delta)$.

We use the following notion of representation divergence.

**Definition B.1** (divergence between two representations). *Given a distribution $q$ over $\mathbb{R}^d$ and two representation functions $\phi, \phi' \in \Phi$, the divergence between $\phi$ and $\phi'$ with respect to $q$ is defined as*

$$D_q (\phi, \phi') = \Sigma_q (\phi', \phi') - \Sigma_q (\phi', \phi) (\Sigma_q (\phi, \phi))^\dagger \Sigma_q (\phi, \phi') \in \mathbb{R}^{k \times k}.$$

It is easy to verify $D_q(\phi, \phi') \succeq 0$, $D_q(\phi, \phi) = 0$ for any $\phi, \phi'$ and $q$. See Lemma B.2's proof.

The next lemma shows a relation between (symmetric) covariance and divergence.

**Lemma B.2.** *Suppose that two representation functions $\phi, \phi' \in \Phi$ and two distributions $q, q'$ over $\mathbb{R}^d$ satisfy $\Lambda_q(\phi, \phi') \succeq \alpha \cdot \Lambda_{q'}(\phi, \phi')$ for some $\alpha > 0$. Then it must hold that $D_q(\phi, \phi') \succeq \alpha \cdot D_{q'}(\phi, \phi')$.*

*Proof.* Fix any $\boldsymbol{v} \in \mathbb{R}^k$. We will prove $\boldsymbol{v}^\top D_q(\phi, \phi')\boldsymbol{v} \geq \alpha \cdot \boldsymbol{v}^\top D_{q'}(\phi, \phi')\boldsymbol{v}$, which will complete the proof of the lemma.

We define a quadratic function $f : \mathbb{R}^k \to \mathbb{R}$ as $f(\boldsymbol{w}) = [\boldsymbol{w}^\top, -\boldsymbol{v}^\top]\Lambda_q(\phi, \phi') \begin{bmatrix} \boldsymbol{w} \\ -\boldsymbol{v} \end{bmatrix}$. According to Definition 5.2, we can write

$$f(\boldsymbol{w}) = \boldsymbol{w}^\top \Sigma_q(\phi, \phi)\boldsymbol{w} - 2\boldsymbol{w}^\top \Sigma_q(\phi, \phi')\boldsymbol{v} + \boldsymbol{v}^\top \Sigma_q(\phi', \phi')\boldsymbol{v}$$
$$= \mathbb{E}_{\boldsymbol{x} \sim q} \left[ \left( \boldsymbol{w}^\top \phi(\boldsymbol{x}) - \boldsymbol{v}^\top \phi'(\boldsymbol{x}) \right)^2 \right].$$

Therefore we have $f(\boldsymbol{w}) \geq 0$ for any $\boldsymbol{w} \in \mathbb{R}^k$.[6] This means that $f$ must have a global minimizer in $\mathbb{R}^k$. Since $f$ is convex, taking its gradient $\nabla f(\boldsymbol{w}) = 2\Sigma_q(\phi, \phi)\boldsymbol{w} - 2\Sigma_q(\phi, \phi')\boldsymbol{v}$ and setting the gradient to $\mathbf{0}$, we obtain a global minimzer $\boldsymbol{w}^* = (\Sigma_q(\phi, \phi))^\dagger \Sigma_q(\phi, \phi')\boldsymbol{v}$. Plugging this into the definition of $f$, we obtain[7]

$$\min_{\boldsymbol{w} \in \mathbb{R}^k} f(\boldsymbol{w}) = f(\boldsymbol{w}^*) = \boldsymbol{v}^\top D_q(\phi, \phi')\boldsymbol{v}. \tag{31}$$

Similarly, letting $g(\boldsymbol{w}) = [\boldsymbol{w}^\top, -\boldsymbol{v}^\top]\Lambda_{q'}(\phi, \phi') \begin{bmatrix} \boldsymbol{w} \\ -\boldsymbol{v} \end{bmatrix}$, we have

$$\min_{\boldsymbol{w} \in \mathbb{R}^k} g(\boldsymbol{w}) = \boldsymbol{v}^\top D_{q'}(\phi, \phi')\boldsymbol{v}.$$

---

[6] Note that we have proved $\Lambda_q(\phi, \phi') \succeq 0$.

[7] Note that (31) implies $D_q(\phi, \phi') \succeq 0$.

From $\Lambda_q(\phi, \phi') \succeq \alpha \cdot \Lambda_{q'}(\phi, \phi')$ we know $f(\boldsymbol{w}) \geq \alpha g(\boldsymbol{w})$ for any $\boldsymbol{w} \in \mathbb{R}^k$. Recall that $\boldsymbol{w}^* \in \arg\min_{\boldsymbol{w} \in \mathbb{R}^k} f(\boldsymbol{w})$. We have

$$\alpha \boldsymbol{v}^\top D_{q'}(\phi, \phi') \boldsymbol{v} = \alpha \min_{\boldsymbol{w} \in \mathbb{R}^k} g(\boldsymbol{w}) \leq \alpha g(\boldsymbol{w}^*) \leq f(\boldsymbol{w}^*)$$
$$= \min_{\boldsymbol{w} \in \mathbb{R}^k} f(\boldsymbol{w}) = \boldsymbol{v}^\top D_q(\phi, \phi') \boldsymbol{v}.$$

This finishes the proof. $\qquad \square$

**Claim B.3** (analogue of Claim A.4). *Under the setting of Theorem 5.1, with probability at least $1 - \delta$ we have*

$$\frac{1}{n_2} \left\| P_{\hat{\phi}(X_{T+1})}^{\perp} \phi^*(X_{T+1}) \right\|_F^2 \lesssim \frac{\sigma^2 \left( \mathcal{G}(\mathcal{F}_{\mathcal{X}}(\Phi))^2 + \log \frac{1}{\delta} \right)}{n_1 \sigma_k^2(W^*)}.$$

*Proof.* We continue to use the notation from Claim B.1 and its proof.

Let $\hat{p}_t$ be the empirical distribution over the samples in $X_t$ ($t \in [T+1]$). According to Assumptions 5.1 and 5.2 as well as the setting in Theorem 5.1, we know that the followings are satisfied with probability at least $1 - \delta$:

$$0.9\Lambda_p(\phi, \phi') \preceq \Lambda_{\hat{p}_t}(\phi, \phi') \preceq 1.1\Lambda_p(\phi, \phi'), \quad \forall \phi, \phi' \in \Phi, \forall t \in [T],$$
$$0.9\Lambda_p(\hat{\phi}, \phi^*) \preceq \Lambda_{\hat{p}_{T+1}}(\hat{\phi}, \phi^*) \preceq 1.1\Lambda_p(\hat{\phi}, \phi^*). \tag{32}$$

Notice that $\hat{\phi}$ and $\phi^*$ are independent of the samples from the target task, so $n_2 \geq N_{\text{point}}(\Phi, p, \frac{\delta}{3})$ is sufficient for the second inequality above to hold with high probability. Using Lemma B.2, we know that (32) implies

$$0.9 D_p(\phi, \phi') \preceq D_{\hat{p}_t}(\phi, \phi') \preceq 1.1 D_p(\phi, \phi'), \quad \forall \phi, \phi' \in \Phi, \forall t \in [T],$$
$$0.9 D_p(\hat{\phi}, \phi^*) \preceq D_{\hat{p}_{T+1}}(\hat{\phi}, \phi^*) \preceq 1.1 D_p(\hat{\phi}, \phi^*). \tag{33}$$

By the optimality of $\hat{\phi}$ and $\hat{\boldsymbol{w}}_1, \ldots, \hat{\boldsymbol{w}}_T$ for (2), we know $\hat{\phi}(X_t)\hat{\boldsymbol{w}}_t = P_{\hat{\phi}(X_t)} \boldsymbol{y}_t = P_{\hat{\phi}(X_t)}(\phi^*(X_t)\boldsymbol{w}_t^* + \boldsymbol{z}_t)$. Then we have the following chain of inequalities:

$$\sigma^2 \left( \mathcal{G}(\mathcal{F}_{\mathcal{X}}(\Phi))^2 + \log \frac{1}{\delta} \right)$$

$$\gtrsim \sum_{t=1}^T \left\| \hat{\phi}(X_t)\hat{\boldsymbol{w}}_t - \phi^*(X_t)\boldsymbol{w}_t^* \right\|^2 \qquad \text{(Claim B.1)}$$

$$= \sum_{t=1}^T \left\| P_{\hat{\phi}(X_t)}(\phi^*(X_t)\boldsymbol{w}_t^* + \boldsymbol{z}_t) - \phi^*(X_t)\boldsymbol{w}_t^* \right\|^2$$

$$= \sum_{t=1}^T \left\| -P_{\hat{\phi}(X_t)}^{\perp} \phi^*(X_t)\boldsymbol{w}_t^* + P_{\hat{\phi}(X_t)}\boldsymbol{z}_t \right\|^2$$

$$= \sum_{t=1}^T \left( \left\| P_{\hat{\phi}(X_t)}^{\perp} \phi^*(X_t)\boldsymbol{w}_t^* \right\|^2 + \left\| P_{\hat{\phi}(X_t)}\boldsymbol{z}_t \right\|^2 \right) \qquad \text{(cross term is 0)}$$

$$\geq \sum_{t=1}^T \left\| P_{\hat{\phi}(X_t)}^{\perp} \phi^*(X_t)\boldsymbol{w}_t^* \right\|^2$$

$$= \sum_{t=1}^T (\boldsymbol{w}_t^*)^\top \phi^*(X_t)^\top \left( I - \hat{\phi}(X_t) \left( \hat{\phi}(X_t)^\top \hat{\phi}(X_t) \right)^{\dagger} \hat{\phi}(X_t)^\top \right) \phi^*(X_t)\boldsymbol{w}_t^*$$

$$= n_1 \sum_{t=1}^T (\boldsymbol{w}_t^*)^\top D_{\hat{p}_t}(\hat{\phi}, \phi^*)\boldsymbol{w}_t^*$$

$$\geq 0.9 n_1 \sum_{t=1}^{T} (\boldsymbol{w}_t^*)^\top D_p(\hat{\phi}, \phi^*) \boldsymbol{w}_t^* \tag{(33)}$$

$$= 0.9 n_1 \left\| \left( D_p(\hat{\phi}, \phi^*) \right)^{1/2} W^* \right\|_F^2$$

$$\geq 0.9 n_1 \left\| \left( D_p(\hat{\phi}, \phi^*) \right)^{1/2} \right\|_F^2 \sigma_k^2(W^*)$$

$$= 0.9 n_1 \mathrm{Tr} \left[ D_p(\hat{\phi}, \phi^*) \right] \sigma_k^2(W^*)$$

$$\geq \frac{0.9 n_1}{1.1} \mathrm{Tr} \left[ D_{\hat{p}_{T+1}}(\hat{\phi}, \phi^*) \right] \sigma_k^2(W^*) \tag{(33)}$$

$$= \frac{0.9 n_1}{1.1 n_2} \left\| P_{\hat{\phi}(X_{T+1})}^\perp \phi^*(X_{T+1}) \right\|_F^2 \sigma_k^2(W^*),$$

completing the proof. $\qquad \square$

Now we can finish the proof of Theorem 5.1.

*Proof of Theorem 5.1.* The excess risk is bounded as

$$\mathrm{ER}(\hat{\phi}, \hat{\boldsymbol{w}}_{T+1})$$

$$= \frac{1}{2} \mathbb{E}_{\boldsymbol{x} \sim p} \left[ \left( \hat{\boldsymbol{w}}_{T+1}^\top \hat{\phi}(\boldsymbol{x}) - (\boldsymbol{w}_{T+1}^*)^\top \phi^*(\boldsymbol{x}) \right)^2 \right]$$

$$= \frac{1}{2} \begin{bmatrix} \hat{\boldsymbol{w}}_{T+1} \\ -\boldsymbol{w}_{T+1}^* \end{bmatrix}^\top \Lambda_p(\hat{\phi}, \phi^*) \begin{bmatrix} \hat{\boldsymbol{w}}_{T+1} \\ -\boldsymbol{w}_{T+1}^* \end{bmatrix}$$

$$\lesssim \begin{bmatrix} \hat{\boldsymbol{w}}_{T+1} \\ -\boldsymbol{w}_{T+1}^* \end{bmatrix}^\top \Lambda_{\hat{p}_{T+1}}(\hat{\phi}, \phi^*) \begin{bmatrix} \hat{\boldsymbol{w}}_{T+1} \\ -\boldsymbol{w}_{T+1}^* \end{bmatrix} \tag{(32)}$$

$$= \frac{1}{n_2} \left\| \hat{\phi}(X_{T+1}) \hat{\boldsymbol{w}}_{T+1} - \phi^*(X_{T+1}) \boldsymbol{w}_{T+1}^* \right\|^2$$

$$= \frac{1}{n_2} \left\| -P_{\hat{\phi}(X_{T+1})}^\perp \phi^*(X_{T+1}) \boldsymbol{w}_{T+1}^* + P_{\hat{\phi}(X_{T+1})} \boldsymbol{z}_{T+1} \right\|^2$$

$$= \frac{1}{n_2} \left( \left\| P_{\hat{\phi}(X_{T+1})}^\perp \phi^*(X_{T+1}) \boldsymbol{w}_{T+1}^* \right\|^2 + \left\| P_{\hat{\phi}(X_{T+1})} \boldsymbol{z}_{T+1} \right\|^2 \right)$$

$$\lesssim \frac{1}{n_2} \left\| P_{\hat{\phi}(X_{T+1})}^\perp \phi^*(X_{T+1}) \boldsymbol{w}_{T+1}^* \right\|^2 + \frac{\sigma^2(k + \log \frac{1}{\delta})}{n_2}. \qquad \text{(using } \chi^2 \text{ tail bound)}$$

Taking expectation over $\boldsymbol{w}_{T+1}^* \sim \nu$, we get

$$\mathbb{E}_{\boldsymbol{w}_{T+1}^* \sim \nu}[\mathrm{ER}(\hat{\phi}, \hat{\boldsymbol{w}}_{T+1})]$$

$$\lesssim \frac{1}{n_2} \left\| P_{\hat{\phi}(X_{T+1})}^\perp \phi^*(X_{T+1}) \right\|_F^2 \left\| \mathbb{E}_{\boldsymbol{w} \sim \nu}[\boldsymbol{w} \boldsymbol{w}^\top] \right\| + \frac{\sigma^2(k + \log \frac{1}{\delta})}{n_2}$$

$$\lesssim \frac{1}{k n_2} \left\| P_{\hat{\phi}(X_{T+1})}^\perp \phi^*(X_{T+1}) \right\|_F^2 + \frac{\sigma^2(k + \log \frac{1}{\delta})}{n_2}$$

$$\lesssim \frac{1}{k} \cdot \frac{\sigma^2 \left( \mathcal{G}(\mathcal{F}_{\mathcal{X}}(\Phi))^2 + \log \frac{1}{\delta} \right)}{n_1 \sigma_k^2(W^*)} + \frac{\sigma^2(k + \log \frac{1}{\delta})}{n_2} \qquad \text{(Claim B.3)}$$

$$\lesssim \frac{\sigma^2 \left( \mathcal{G}(\mathcal{F}_{\mathcal{X}}(\Phi))^2 + \log \frac{1}{\delta} \right)}{n_1 T} + \frac{\sigma^2(k + \log \frac{1}{\delta})}{n_2}, \qquad (\sigma_k(W^*) \gtrsim \frac{T}{k})$$

finishing the proof. $\qquad \square$

# C  PROOF OF THEOREM 6.1

## C.1  PROOF SKETCH OF THEOREM 6.1

Let $R = \|\Theta^*\|_*$. Recall $\hat{B}$ and $\hat{W}$ are derived from Eqn. (12) and let $\hat{\Theta} := \hat{B}\hat{W}$. We first note that the constraint set $\{\|\boldsymbol{w}\|_i^2 \leq R/T, \|B\|_F^2 \leq R\}$ ensures $\|W\|_F^2 \leq R$ and $\|WB\|_* \leq R$ at global minimum. On the other hand, our constraint for $W, B$ is also expressive enough to attain any $\hat{\Theta}$ that satisfies $\|\hat{\Theta}\|_* \leq R$. See reference e.g. Srebro and Shraibman (2005). Therefore at global minimum $\|\hat{W}\|_F \leq \sqrt{R}, \|\hat{B}\|_F \leq \sqrt{R}$ and $\|\hat{\Theta}\|_* \leq R$.

For the ease of proof, we introduce the following auxiliary functions and parameters. Write

$$L_1(W) = \frac{1}{2} \left\| \Sigma^{1/2}\Theta^* - \Sigma^{1/2}\hat{B}W \right\|_F^2,$$

$$L_1^\lambda(W) = L_1(W) + \frac{\lambda}{2}\|W\|_F^2, \qquad\qquad W_1^\lambda \leftarrow \arg\min_W \{L_1^\lambda(W)\},$$

$$L_2(\boldsymbol{w}) = \frac{1}{2} \left\| \Sigma^{1/2}\boldsymbol{\theta}_{T+1}^* - \Sigma^{1/2}\hat{B}\boldsymbol{w} \right\|^2, \qquad \boldsymbol{w}_2^\lambda \leftarrow \arg\min_{\boldsymbol{w}} \{L_2(\boldsymbol{w}) + \lambda/2\|w\|^2\},$$

$$\hat{L}_1(W) = \frac{1}{2n_1} \left\| \mathcal{X}(\Theta^* - \hat{B}W) \right\|^2,$$

$$\hat{L}_1^\lambda(W) = \hat{L}_1(W) + \frac{\lambda}{2}\|W\|_F^2 \qquad\qquad \bar{W}_1^\lambda \leftarrow \arg\min_W \{\hat{L}_1^\lambda(W)\},$$

$$\hat{L}_2(\boldsymbol{w}) = \frac{1}{2n_2} \left\| X_{T+1}\boldsymbol{\theta}_{T+1}^* - X_{T+1}\hat{B}\boldsymbol{w} \right\|^2, \qquad \bar{\boldsymbol{w}}_2 \leftarrow \arg\min_{\boldsymbol{w} \leq r} \{\hat{L}_2(\boldsymbol{w})\}.$$

We define terms $\epsilon_{ic,1}$ and $\epsilon_{ic,2}$ that will be used to bound intrinsic dimension concentration error in the input signal. Namely with high probability, $\|\Sigma^{1/2}\Theta\| - \sqrt{1/n_1 \sum_{t=1}^T \|X_t\theta_t\|^2} \leq \epsilon_{ic,1}\|\Theta\|_*$, and similarly $\|\Sigma^{1/2}\hat{B}\boldsymbol{v}\| - \sqrt{\frac{1}{n_2}\|X\hat{B}\boldsymbol{v}\|^2} \leq \epsilon_{ic,2}\|v\|_2$. Additionally we use $\epsilon_{ee,i}, i \in \{1,2\}$ to bound the estimation error (for fixed design) incurred when using noisy label $\boldsymbol{y}_{T+1}$ and $Y$.

The choice of $\epsilon_{ee,i}$, and $\epsilon_{ic,i}$ are respectively justified in Lemma C.5, Claim C.4, Lemma C.10 and Claim C.11, along with some more detailed descriptions.

*Proof of Theorem 6.1.*

$$\mathbb{E}_{\boldsymbol{\theta}^* \sim \nu} ER(\hat{B}, \hat{\boldsymbol{w}}_{T+1})$$

$$= \mathbb{E}_{\boldsymbol{\theta}^* \sim \nu} L_2(\hat{\boldsymbol{w}}_{T+1})$$

$$\lesssim \mathbb{E}_{\boldsymbol{\theta}^* \sim \nu} \hat{L}_2(\hat{\boldsymbol{w}}_{T+1}) + \epsilon_{ic,2}^2 r^2 \qquad\qquad \text{(Claim C.11)}$$

$$\lesssim \mathbb{E}_{\boldsymbol{\theta}^* \sim \nu} \hat{L}_2(\bar{\boldsymbol{w}}_2) + \epsilon_{ee,2}^2 r + \epsilon_{ic,2}^2 r^2 \qquad\qquad \text{(Lemma C.4)}$$

$$\leq \mathbb{E}_{\boldsymbol{\theta}^* \sim \nu} \hat{L}_2(\boldsymbol{w}_2^\lambda) + \epsilon_{ee,2}^2 r + \epsilon_{ic,2}^2 r^2 \qquad\qquad \text{(Definition of } \bar{\boldsymbol{w}}_2)$$

$$\lesssim \mathbb{E}_{\boldsymbol{\theta}^* \sim \nu} L_2(\boldsymbol{w}_2^\lambda) + \epsilon_{ee,2}^2 r + \epsilon_{ic,2}^2 r^2 \qquad\qquad \text{(Claim C.11)}$$

$$= \frac{1}{T} L_1(W_1^\lambda) + \epsilon_{ee,2}^2 r + \epsilon_{ic,2}^2 r^2 \qquad\qquad \text{(Claim C.3)}$$

$$\lesssim \frac{\lambda R}{T} + \epsilon_{ee,2}^2 r + \epsilon_{ic,2}^2 r^2 \qquad\qquad \text{(Lemma C.2)}$$

$$\lesssim \frac{\epsilon_{ee,1}^2 R + \epsilon_{ic,1}^2 R^2}{T} + \epsilon_{ee,2}^2 r + \epsilon_{ic,2}^2 r^2. \qquad\qquad \text{(Choices of } \lambda)$$

Each step is with high probability $1 - \delta/10$ over the randomness of $\mathcal{X}$ or $X_{T+1}$. Therefore overall by union bound, with probability $1 - \delta$, by plugging in the values of $\epsilon_{ic,i}$ and $\epsilon_{ee,i}$ we have:

$$\mathbb{E}_{\boldsymbol{\theta}^* \sim \nu} ER(\hat{B}, \hat{\boldsymbol{w}}_{T+1}) \leq \frac{\sigma R}{\sqrt{T}} \tilde{O}\left( \frac{\sqrt{\text{Tr}(\Sigma)}}{\sqrt{Tn_1}} + \frac{\sqrt{\|\Sigma\|}}{\sqrt{n_2}} \right) + \frac{\rho^4 R^2}{T} \tilde{O}\left( \frac{\text{Tr}\Sigma}{n_1} + \frac{\|\Sigma\|}{n_2} \right).$$

Notice a term $\|\Sigma\|/n_1$ is absorbed by $\|\Sigma\|/n_2$ since we assume $n_1 \geq n_2$. $\qquad\qquad\square$

**Claim C.1** (guarantee with source regularization).

$$\frac{1}{n_1}\|\mathcal{X}(\Theta^* - \hat{\Theta})\|_F^2 + \lambda\|\hat{\Theta}\|_* \leq 3\lambda\|\Theta^*\|_* \leq 3\lambda R,$$

and $\|\hat{B}\|_F^2 \leq 3R$, $\|\hat{W}\|_F^2 \leq 3R$ for any $\lambda \geq \frac{2}{n}\|\mathcal{X}^*(Z)\|_2$.

Here $\mathcal{X}^*$ is the adjoint operator of $\mathcal{X}$ such that $\mathcal{X}^*(Z) = \sum_{i=1}^T X_t^\top z_t e_t^\top$.

*Proof.* With the optimality of $\hat{\Theta}$ we have:

$$\frac{1}{2n_1}\|\mathcal{X}(\hat{\Theta} - \Theta^*) - Z\|_F^2 + \lambda\|\hat{\Theta}\|_* \leq \frac{1}{2n_1}\|Z\|_F^2 + \lambda\|\Theta^*\|_*,$$

Let $\Delta = \hat{\Theta} - \Theta^*$. Therefore

$$\frac{1}{2n_1}\|\mathcal{X}(\Delta)\|_F^2$$

$$\leq \lambda(\|\Theta^*\|_* - \|\hat{\Theta}\|_*) + \frac{1}{n_1}\langle\Delta, \mathcal{X}^*(Z)\rangle$$

$$\leq \lambda\|\Theta^*\|_* + \frac{1}{n_1}\|\Theta^*\|_* \cdot \|\mathcal{X}^*(Z)\| + \frac{1}{n_1}\|\hat{\Theta}\|_* \cdot \|\mathcal{X}^*(Z)\| - \lambda\|\hat{\Theta}\|_*$$

$$\leq \lambda\|\Theta^*\|_* + \lambda/2\|\Theta^*\|_* + \lambda/2\|\hat{\Theta}\|_* - \lambda\|\hat{\Theta}\|_*$$

$$\text{(Let } \lambda \geq \frac{2}{n_1}\|\mathcal{X}^*(Z)\|)$$

$$= \frac{3}{2}\lambda\|\Theta^*\|_* - \frac{1}{2}\lambda\|\hat{\Theta}\|_*.$$

Therefore $\frac{1}{2n_1}\|\mathcal{X}(\Delta)\|_F^2 + \frac{\lambda}{2}\|\hat{\Theta}\|_* \leq \frac{3}{2}\lambda\|\Theta^*\|_*$, and clearly both terms satisfy $\frac{1}{n_1}\|\mathcal{X}(\Delta)\|_F^2 \leq 3\lambda\|\Theta^*\|_*$ and $\|\hat{\Theta}\|_* \leq 3\|\Theta^*\|_*$.

$\square$

**Lemma C.2** (source task concentration). *For a fixed $\delta > 0$, let $\lambda = \epsilon_{ee,1}^2 + \epsilon_{ic,1}^2 R$, we have*

$$L_1^\lambda(W_1^\lambda) \lesssim \lambda R$$
$$\|W_1^\lambda\|_F \lesssim \sqrt{R}.$$

*with probability $1 - \delta/10$.*

*Proof of Lemma C.2.*

$$\|W_1^\lambda\|_F^2 < \frac{2}{\lambda}L_1^\lambda(W_1^\lambda)$$

$$\leq \frac{2}{\lambda}L_1^\lambda(\hat{W}) \qquad\qquad \text{(Definition of } W_1^\lambda)$$

$$= \frac{2}{\lambda}\left\{\frac{1}{2}\|\Sigma^{1/2}(\Theta^* - \hat{\Theta})\|_F^2 + \frac{\lambda}{2}\|\hat{W}\|_F^2\right\}$$

$$\leq \frac{2}{\lambda}\left\{\frac{1}{2}(\frac{1}{\sqrt{n_1}}\|\mathcal{X}(\Theta^* - \hat{\Theta})\|_F + O(\epsilon_{ic,1})R)^2 + \frac{\lambda}{2}\|\hat{W}\|_F^2\right\}$$

$$\leq \frac{2}{\lambda}\left\{\frac{1}{n_1}\|\mathcal{X}(\Theta^* - B\bar{W}^\lambda)\|_F^2 + \frac{\lambda}{2}\|\bar{W}^\lambda\|_F^2 + O(\epsilon_{ic,1}^2 R^2)\right\}$$

$$\leq \frac{2}{\lambda}\left\{6\lambda R + O(\epsilon_{ic,1}^2 R^2)\right\} \qquad\qquad \text{(from Claim C.1)}$$

$$= \frac{2}{\lambda}O(\lambda R)$$

$$= O(R).$$

Thus both results have been shown. $\square$

**Claim C.3** (Source and Target Connections)**.**

$$\mathbb{E}_{\boldsymbol{\theta}^* \sim \nu} L_2(\boldsymbol{w}_2^\lambda) = L_1(W_1^\lambda)$$

*Proof of Claim C.3.*

$$\boldsymbol{w}_2^\lambda = (\hat{B}^\top \Sigma \hat{B} + \lambda I)^{-1} \hat{B}^\top \Sigma \boldsymbol{\theta}_{T+1}^* =: S_\lambda \boldsymbol{\theta}_{T+1}^*,$$

where $S_\lambda := (\hat{B}^\top \Sigma \hat{B} + \lambda I)^{-1} \hat{B}^\top \Sigma$.

$$\begin{aligned}
\mathbb{E}_{\boldsymbol{\theta}^* \sim \nu} L_2(\boldsymbol{w}_2^\lambda) &= \mathbb{E}_{\boldsymbol{\theta}^* \sim \nu} \|\Sigma^2 (I - S_\lambda) \boldsymbol{\theta}_{T+1}^*\|^2 \\
&= \frac{1}{T} \|\Sigma^2 (I - S_\lambda) \Theta_{T+1}^*\|^2 \\
&= \frac{1}{T} L_1(W^\lambda).
\end{aligned}$$

$\square$

**Lemma C.4** (Estimation Error for Target Task)**.**

$$\hat{L}_2(\hat{\boldsymbol{w}}) - \hat{L}_2(\bar{\boldsymbol{w}})$$
$$\leq \frac{R}{\sqrt{T n_2}} \sigma (\log 1/\delta)^{3/2} \log(n_2) \sqrt{\|\Sigma\|} =: \epsilon_{ee,2}^2 r.$$

*Proof of Lemma C.4.* With the definition of $\hat{\boldsymbol{w}}$ we write the basic inequality:

$$\frac{1}{2n_2} \|X_{T+1}(\hat{B}\hat{\boldsymbol{w}} - \boldsymbol{\theta}^*) - \boldsymbol{z}_{T+1}\|_F^2 \leq \frac{1}{2n_2} \|X_{T+1}(\hat{B}\bar{\boldsymbol{w}} - \boldsymbol{\theta}^*) - \boldsymbol{z}_{T+1}\|_F^2,$$

Therefore by rearranging we get:

$$\begin{aligned}
\frac{1}{2n_2} \|X_{T+1}\hat{B}(\hat{\boldsymbol{w}} - \bar{\boldsymbol{w}})\|_F^2 &\leq \frac{1}{n_2} \langle \hat{\boldsymbol{w}} - \bar{\boldsymbol{w}}, \hat{B}^\top X_{T+1}^\top \boldsymbol{z}_{T+1} \rangle \\
&\leq \frac{\sqrt{R/T}}{n_2} \|\hat{B}^\top X_{T+1}^\top \boldsymbol{z}_{T+1}\|_F^2 \\
&\lesssim \frac{R}{\sqrt{T n_2}} \sigma \log^{2/3}(1/\delta) \log(n_2) \sqrt{\|\Sigma\|} \qquad \text{(Claim C.6)} \\
&= O(r \epsilon_{ee,2}^2)
\end{aligned}$$

$\square$

## C.2 TECHNICAL LEMMAS

This section includes the technical details for several parts: bounding the noise term from basic inequality; and intrinsic dimension concentration for both source and target tasks.

**Lemma C.5** (Regularizer Estimation)**.** *For $X \in \mathbb{R}^{n \times d}$ drawn from distribution $p$ with covariance matrix $\Sigma$, and noise $Z \sim \mathcal{N}(0, \sigma^2 I_n)$, with high probability $1 - \delta$, we have*

$$\epsilon_{ee,1}^2 := \frac{1}{n} \|X^\top Z\|_2 \leq \frac{1}{\sqrt{n}} \sigma \left( \log \frac{1}{\delta} \right)^{3/2} \log(T + n) \sqrt{T \|\Sigma\| + \text{Tr}(\Sigma)}.$$

*Proof.* We use matrix Bernstein with intrinsic dimension to bound $\lambda$ (See Theorem 7.3.1 in Tropp et al. (2015)).

Write $A = \frac{1}{\sqrt{n}} X^\top Z = \frac{1}{\sqrt{n}} \sum_{t=1}^T X^\top \boldsymbol{z}_t \boldsymbol{e}_t^\top =: \sum_{t=1}^T S_t$.

$$\begin{aligned}
\mathbb{E}_{X,Z}[AA^\top] &= \mathbb{E}_X \left[ \sum_{t=1}^T \frac{1}{n_1} X^\top \mathbb{E}_Z[\boldsymbol{z}_t \boldsymbol{z}_t^\top] X \right] \\
&= \sigma^2 T \Sigma
\end{aligned}$$

$$\mathbb{E}_{X,Z}[A^\top A] = \sum_{t=1}^{T} \frac{1}{n} e\, \mathbb{E}_{X,Z}\left[z_t^\top X X^\top z_t e_t^\top\right]$$

$$= \sum_{t=1}^{T} \frac{1}{n} \mathbb{E}_{X,Z}[z_t^\top X X^\top z_t] e_t e_t^\top$$

$$= \sigma^2 \mathrm{Tr}(\Sigma) I_n.$$

Therefore the matrix variance statistic of the sum $v(A)$ satisfies: $v(A) = \sigma^2 \max\{T\|\Sigma\|, \mathrm{Tr}(\Sigma)\}$. Denote $V = \mathrm{diag}([T\Sigma, \mathrm{Tr}(\Sigma)I])$ and its intrinsic dimension $d_\Sigma = \mathrm{tr}(V)/\|V\|$. $\mathrm{Tr}(V) = \sigma^2(T + n)\mathrm{Tr}(\Sigma)$, and $\|V\|_2 \geq \sigma^2 \mathrm{Tr}(\Sigma)$. Therefore $d_\Sigma \leq T + n$.

Finally from Hanson-Wright inequality, the upper bound on each term is $\|S_t\|^2 \leq \|Xz_t\|^2 \leq \sigma^2 \mathrm{Tr}(\Sigma) + \sigma^2\|\Sigma\|\log\frac{1}{\delta} + \sigma^2\|\Sigma\|_F\sqrt{\log\frac{1}{\delta}}$ with probability $1 - \delta$. Thus using $\|\Sigma\|_F \leq \mathrm{Tr}(\Sigma)$,

$$\|S_t\| \leq \sigma\sqrt{(1 + \sqrt{\log\frac{1}{\delta}})\mathrm{Tr}(\Sigma) + \|\Sigma\|\log\frac{1}{\delta}} =: L.$$

Then from intrinsic matrix bernstein (Theorem 7.3.1 in Tropp et al. (2015)), with probability $1 - \delta$ we have, $\|A\| \leq \mathcal{O}(\sigma\sqrt{\log\frac{1}{\delta}v\log(d_\Sigma)} + \sigma\log\frac{1}{\delta}L\log(d_\Sigma))$, which gives

$$\|A\| \leq \sigma\sqrt{\log\frac{1}{\delta}T\|\Sigma\|\log(T + n) + \log\frac{1}{\delta}\mathrm{Tr}(\Sigma)\log(T + n)} + \log\frac{1}{\delta}\sigma L\log(T + n)$$

$$\lesssim \sigma\left(\log\frac{1}{\delta}\right)^{3/2}\log(T + n)\sqrt{T\|\Sigma\| + \mathrm{Tr}(\Sigma)}.$$

$\square$

**Claim C.6** (target noise concentration). *For a fixed $\delta > 0$, with probability $1 - \delta/10$, $\epsilon_{ee,2}^2 :=$* $\frac{1}{\sqrt{n_2}}\|\hat{B}^\top X_{T+1}^\top z\|_2 \leq O(\log^{2/3}(1/\delta)\log(n_2)\sqrt{\mathrm{Tr}(\hat{B}^\top\Sigma\hat{B})}) \leq \tilde{\mathcal{O}}(\sqrt{\|\Sigma\|_2 R})$.

*Proof.* The first inequality directly follows from Lemma C.5. Meanwhile $\mathrm{Tr}(\hat{B}^\top\Sigma\hat{B}) = \langle\Sigma, \hat{B}\hat{B}^\top\rangle \leq \|\Sigma\|_2\|\hat{B}\hat{B}^\top\|_* \lesssim \|\Sigma\|_2 R$. This finishes the proof. $\square$

**Definition C.7.** *The sub-gaussian norm of some vector $y$ is defined as:*

$$\|y\|_{\psi_2} := \sup_{x\in\mathbb{S}^{n-1}}\|\langle y, x\rangle\|_{\psi_2}, \tag{34}$$

*where $\mathbb{S}^{n-1}$ denotes the unit Euclidean sphere in $\mathbb{R}^n$.*

**Definition C.8.** *Let $T \subset \mathbb{R}^d$ be a bounded set, and $g$ be a standard normal random vector in $\mathbb{R}^d$, i.e., $g \sim \mathcal{N}(0, I_d)$. Then the quantities*

$$w(T) := \mathbb{E}\sup_{x\in T}\langle g, x\rangle, \text{ and } \gamma(T) := \mathbb{E}\sup_{x\in T}|\langle g, x\rangle| \tag{35}$$

*are called the Gaussian width of $T$ and the Gaussian complexity of $T$, respectively.*

**Theorem C.9** (Restated Matrix deviation inequality from Vershynin (2017)). *Let $A$ be an $m \times n$ matrix whose rows $a_i$ are independent, isotropic and sub-gaussian random vectors in $\mathbb{R}^n$. Let $T \subset \mathbb{R}^n$ be a fixed bounded set. Then*

$$\mathbb{E}\sup_{x\in T}|\|Ax\|_2 - \sqrt{m}\|x\|_2| \leq C\rho^2\gamma(T), \tag{36}$$

*where $K = \max_i\|A_i\|_{\psi_2}$ is the maximal sub-gaussian norm of the rows of $A$. A high-probability version states as follows. With probability $1 - \delta$,*

$$\sup_{x\in T}|\|Ax\|_2 - \sqrt{m}\|x\|_2| \leq C\rho^2[\gamma(T) + \sqrt{\log(2/\delta)}r(T)], \tag{37}$$

*where the radius $r(T) := \sup_{x\in T}\|x\|_2$.*

**Lemma C.10** (intrinsic dimension concentration). *Let $X, X_t, t \in [T]$ be $n \times d$ matrix whose rows $\boldsymbol{x}$ are independent, isotropic and sub-gaussian random vectors in $\mathbb{R}^d$ that satisfy Assumption 4.1, and the whitening distribution is with sub-gaussian norm $C_1 \rho$, where $\mathbb{E}[\boldsymbol{x}] = 0$ and $\mathbb{E}[\boldsymbol{x}\boldsymbol{x}^\top] = \Sigma$. For a fixed $\delta > 0$, and any $\boldsymbol{v} \in \mathbb{R}^d$, we have*

$$\|\Sigma^{1/2}\boldsymbol{v}\|_2 \leq \frac{1}{\sqrt{n}}\|X\boldsymbol{v}\|_2 + \frac{C\rho^2}{\sqrt{n}}\left(\sqrt{\mathrm{Tr}(\Sigma)} + \sqrt{\log(2/\delta)\|\Sigma\|}\right)\|\boldsymbol{v}\|_2.$$

*For any $\Theta \in \mathbb{R}^{d \times T}$, we further have*

$$\|\Sigma^{1/2}\Theta\|_F \leq \frac{1}{\sqrt{n}}\sqrt{\sum_{t=1}^{T}\|X_t\boldsymbol{\theta}_t\|_2^2} + \epsilon_{ic,1}\|\Theta\|_*, \tag{38}$$

*where $\epsilon_{ic,1} := \frac{2C\rho^2}{\sqrt{n}}\left(\sqrt{\mathrm{Tr}(\Sigma)} + \sqrt{\log(2/\delta)\|\Sigma\|}\right)$, with probability $1 - \delta$.*

*Proof.* We use Theorem C.9. Let $T = \{\boldsymbol{v} : |\Sigma^{-1/2}\boldsymbol{v}|_2 \leq 1\}$. Let $\boldsymbol{x} = \Sigma^{1/2}\boldsymbol{z}, X = Z\Sigma^{1/2}$. Then $\gamma(T) = \sqrt{\mathrm{Tr}(\Sigma)}, r(T) = \|\Sigma\|^{1/2}$. We note with probability $1 - \delta$,

$$\sup_{\|\boldsymbol{v}\|=1}\left|\frac{1}{\sqrt{n}}\|X\boldsymbol{v}\|_2 - \|\Sigma^{1/2}\boldsymbol{v}\|_2\right|$$

$$= \sup_{\bar{\boldsymbol{v}} \in T}\left|\frac{1}{\sqrt{n}}\|Z\bar{\boldsymbol{v}}\|_2 - \|\bar{\boldsymbol{v}}\|_2\right|$$

$$\leq \frac{C\rho^2}{\sqrt{n}}\left(\gamma(T) + \sqrt{\log(2/\delta)}r(T)\right)$$

$$= \frac{C\rho^2}{\sqrt{n}}\left(\sqrt{\mathrm{Tr}(\Sigma)} + \sqrt{\log(2/\delta)\|\Sigma\|}\right).$$

Therefore $\left|\frac{1}{\sqrt{n}}\|X\boldsymbol{v}\|_2 - \|\Sigma^{1/2}\boldsymbol{v}\|_2\right| \leq \frac{C\rho^2}{\sqrt{n}}\left(\sqrt{\mathrm{Tr}(\Sigma)} + \sqrt{\log(2/\delta)\|\Sigma\|}\right), \forall \|\boldsymbol{v}\| = 1$. Then by homogeneity of $\boldsymbol{v}$, for arbitrary $\boldsymbol{v}$, we have

$$\left|\frac{1}{\sqrt{n}}\|X\boldsymbol{v}\|_2 - \|\Sigma^{1/2}\boldsymbol{v}\|_2\right| \leq \|\boldsymbol{v}\|_2 \underbrace{\frac{C\rho^2}{\sqrt{n}}\left(\sqrt{\mathrm{Tr}(\Sigma)} + \sqrt{\log(2/\delta)\|\Sigma\|}\right)}_{\text{term I}}.$$

Notice when $n \gg C^2\rho^4(\mathrm{Tr}(\Sigma) + \|\Sigma\|\log 1/\delta)$, term I $\leq 0.1\sqrt{\lambda}$. Therefore $|\|\Sigma^{1/2}\boldsymbol{v}\|_2 - \frac{1}{\sqrt{n}}\|X\boldsymbol{v}\|_2| \leq 0.1\sqrt{\lambda}\|\boldsymbol{v}\|$.

Write $\Theta = UDV^\top$, where $D = \mathrm{diag}(\sigma_1, \sigma_2, \cdots, \sigma_T)$.

$$\frac{1}{n}\sum_{t=1}^{T}\|X_t\boldsymbol{\theta}_t\|_2^2 = \frac{1}{n}\sum_{t=1}^{T}\sigma_t^2\|X_t\boldsymbol{u}_t\|^2$$

$$\geq \sum_{t=1}^{T}\sigma_t^2\left(\|\Sigma^{1/2}\boldsymbol{u}_t\|_2 - \|\boldsymbol{u}_t\|_2\frac{C\rho^2}{\sqrt{n}}\left(\sqrt{\mathrm{Tr}(\Sigma)} + \sqrt{\log(2/\delta)\|\Sigma\|}\right)\right)^2$$

$$> \sum_{t=1}^{T}\sigma_t^2\left(\|\Sigma^{1/2}\boldsymbol{u}_t\|_2^2 - 2\|\Sigma^{1/2}\boldsymbol{u}_t\|\|\boldsymbol{u}_t\|_2\frac{C\rho^2}{\sqrt{n}}\left(\sqrt{\mathrm{Tr}(\Sigma)} + \sqrt{\log(2/\delta)\|\Sigma\|}\right)\right)$$

$$= \|\Sigma^{1/2}\Theta\|_F^2 - \frac{2C\rho^2}{\sqrt{n}}\left(\sqrt{\mathrm{Tr}(\Sigma)} + \sqrt{\log(2/\delta)\|\Sigma\|}\right)\sum_{t}\sigma_t(\sigma_t\|\Sigma^{1/2}\boldsymbol{u}_t\|_2)$$

$$\geq \|\Sigma^{1/2}\Theta\|_F^2 - \frac{2C\rho^2}{\sqrt{n}}\left(\sqrt{\mathrm{Tr}(\Sigma)} + \sqrt{\log(2/\delta)\|\Sigma\|}\right)\sum_{t}\sigma_t(\max_t\sigma_t\|\Sigma^{1/2}\boldsymbol{u}_t\|_2)$$

$$\geq \|\Sigma^{1/2}\Theta\|_F^2 - \frac{2C\rho^2}{\sqrt{n}}\left(\sqrt{\mathrm{Tr}(\Sigma)} + \sqrt{\log(2/\delta)\|\Sigma\|}\right)\|\Theta\|_*\|\Sigma^{1/2}\Theta\|_F.$$

Therefore $\|\Sigma^{1/2}\Theta\|_F \leq \frac{1}{\sqrt{n}}\sqrt{\sum_{t=1}^{T}|X_t\boldsymbol{\theta}_t\|_2^2} + \underbrace{\frac{2C\rho^2}{\sqrt{n}}\left(\sqrt{\mathrm{Tr}(\Sigma)} + \sqrt{\log(2/\delta)\|\Sigma\|}\right)\|\Theta\|_*}_{\text{term II}}.$

$\square$

**Claim C.11.** *Let $X$ be $n_2 \times d$ matrix whose rows $\boldsymbol{x}$ are independent, isotropic and sub-gaussian random vectors in $\mathbb{R}^d$ that satisfy Assumption 4.1, where $\mathbb{E}[\boldsymbol{x}] = 0$ and $\mathbb{E}[\boldsymbol{x}\boldsymbol{x}^\top] = \Sigma$. Let $\Sigma_B = B^\top\Sigma B$ for some matrix $B$ that satisfies $\|BB^\top\|_* \lesssim R$. Then for a fixed $\delta > 0$, and any $\boldsymbol{v} \in \mathbb{R}^d$ we have: $\|\Sigma_B^{1/2}\boldsymbol{v}\| \leq \frac{1}{n_2}\|X\hat{B}\boldsymbol{v}\| + \epsilon_{ic,2}\|v\|$, where $\epsilon_{ic,2} := \frac{C\rho^2}{\sqrt{n_2}}\sqrt{R\|\Sigma\|\log(1/\delta)}$, and $C$ is a universal constant.*

*Proof.* This result directly uses Lemma C.10 when replacing $X$ by $X\hat{B}$. Notice now the subgaussian norm for the whitening distribution for $\hat{B}^\top\boldsymbol{x}$ remains the same as $C_1\rho$. Therefore $\|\Sigma^{1/2}\hat{B}\boldsymbol{v}\|_2 \leq \frac{1}{\sqrt{n_2}}\|X\hat{B}\boldsymbol{v}\|_2 + S\|\boldsymbol{v}\|_2 \leq \|\Sigma^{1/2}\hat{B}\boldsymbol{v}\|_2 \leq \frac{1}{\sqrt{n_2}}\|X\boldsymbol{v}\|_2 + S\|\boldsymbol{v}\|$. Here $S = C\rho^2/\sqrt{n_2}(\sqrt{\mathrm{Tr}(\Sigma_B)} + \sqrt{\log(2/\delta)\|\Sigma_B\|}) \leq C'\rho^2/\sqrt{n_2}\sqrt{\|\Sigma\|R\log(1/\delta)} =: \epsilon_{ic,2}$.

$\square$

## D    PROOF OF THEOREM 7.1

First, we describe a standard lifting of neural networks to infinite dimension linear regression Wei et al. (2019); Rosset et al. (2007); Bengio et al. (2006). Define the infinite feature vector with coordinates $\phi(\boldsymbol{x})_{\boldsymbol{b}} = (\boldsymbol{b}^\top\boldsymbol{x})_+$ for every $\boldsymbol{b} \in \mathbb{S}^{d_0-1}$. Let $\alpha_t$ be a signed measure on $\mathbb{S}^{d_0-1}$. The inner product notation denotes integration: $\alpha^\top\phi(\boldsymbol{x}) \triangleq \int_{\mathbb{S}^{d_0-1}}\phi(\boldsymbol{x})_{\boldsymbol{b}}d\alpha(\boldsymbol{b})$. The $t^{th}$ output of the infinite-width neural network is $f_{\alpha_t}(\boldsymbol{x}) = \langle\alpha_t, \phi(\boldsymbol{x})\rangle$. Consider the least-squares problem

$$\min_{\substack{\alpha_1,\ldots,\alpha_t:|\mathrm{supp}(\alpha_t)|\leq d \\ \|\bar{\alpha}\|_{2,1}\leq R}} \frac{1}{2n}\sum_{i,t}(y_{it} - \alpha_t^\top\phi(\boldsymbol{x}_i))^2, \tag{39}$$

where $\boldsymbol{\alpha}(u) = [\alpha_1(u),\ldots,\alpha_T(u)]$, and $\|\boldsymbol{\alpha}\|_{2,1} = \int_{\mathbb{S}^{d_0-1}}\|\boldsymbol{\alpha}(\bar{\boldsymbol{b}})\|_2 d(\bar{\boldsymbol{b}})$. The regularizer corresponds to a group $\ell_1$ regularizer on the vector measure $\boldsymbol{\alpha}$.

**Proposition D.1.** *Let $\gamma_d$ be the value of Equation (17) when the network has $d$ neurons and $\gamma_d^\star$ be the value of Equation (39). Then*

$$\gamma_d = \gamma_d^\star. \tag{40}$$

*Proof.* Let $B, W$ be solutions to Equation (17). Let $\bar{B} = BD_\beta^{-1}$ and $D_\beta$ be a diagonal matrix whose entries are $\beta_j = \|B^\top\boldsymbol{e}_j\|_2$. The network $f_{B,W}(\boldsymbol{x}) = W^\top D_\beta(\bar{B}^\top\boldsymbol{x})_+$ and it satisfies

$$\|\beta\|_2 = \|B\|_F.$$

We first show that $\gamma_d^\star \leq \gamma_d$. Define $\alpha_t(\frac{\boldsymbol{b}_j}{\|\boldsymbol{b}_j\|}) = W_{tj}\beta_j$. We verify that

$$\alpha_t^\top\phi(\boldsymbol{x}) = \sum_{j=1}^{d}\alpha_t(\bar{\boldsymbol{b}}_j)\phi(\boldsymbol{x})_{\bar{\boldsymbol{b}}_j} = \sum_{j=1}^{d}W_{tj}\beta_j(\bar{\boldsymbol{b}}_j^\top\boldsymbol{x})_+ = \boldsymbol{w}_t^\top(B^\top\boldsymbol{x})_+ = f_{B,\boldsymbol{w}_t}(\boldsymbol{x}).$$

Due to the regularizer, and using the AM-GM inequality, at optimality $\beta_j = \|W\boldsymbol{e}_j\|_2$. Next, we verify that the two regularizer values are the same. Let $\bar{\boldsymbol{w}}_j$ be the $j$-th row vector of $W$. We have

$$\|\alpha\|_{2,1} = \sum_{j=1}^{d}\beta_j\|\bar{\boldsymbol{w}}_j\|$$

$$\leq \sum_{j=1}^{d}\beta_j^2/2 + \|\bar{\boldsymbol{w}}_j\|^2/2$$

$$= \frac{1}{2}\|\beta\|^2 + \frac{1}{2}\|W\|_F^2 \le R.$$

Thus the network given by $\alpha_t^\top \phi(\boldsymbol{x})$ has the same network outputs and regularizer values. Thus $\gamma^\star \le \gamma_d$.

Finally, we show that $\gamma_d \le \gamma_d^\star$. Let $\bar{\boldsymbol{b}}_j$ for $j \in [d]$ be the support of the optimal measure of (39). Define $\beta_j = \sqrt{\|\boldsymbol{\alpha}(\bar{\boldsymbol{b}}_j)\|_2}$, $B = \bar{B}D_\beta$ where $\bar{B}$ is a matrix whose rows are $\bar{\boldsymbol{b}}_j$, and $W$ such that $W_{jt} = \alpha_t(\bar{\boldsymbol{b}}_j)/\sqrt{\|\boldsymbol{\alpha}(\bar{\boldsymbol{b}}_j)\|}$.

We verify that the network values agree

$$\boldsymbol{e}_t^\top f_{B,W}(\boldsymbol{x}) = \boldsymbol{e}_t^\top W^\top D_\beta(\bar{B}^\top \boldsymbol{x})_+ = \sum_j W_{jt}\beta_j(\bar{\boldsymbol{b}}_j^\top \boldsymbol{x})_+ = \alpha^\top \phi(\boldsymbol{x}).$$

Finally by our construction $\beta_j = \|W\boldsymbol{e}_j\|$, so the regularizer values agree. Thus $\gamma_d = \gamma_d^\star$. $\qquad\square$

Finally, we note that the regularizer can be expressed in a variational form as[8]

$$\|\alpha\|_{2,1} = \min_{\boldsymbol{b},W:\alpha_t(\bar{\boldsymbol{b}})=\beta(\bar{\boldsymbol{b}})\boldsymbol{w}_t(\bar{\boldsymbol{b}})} \|\beta\|_2^2 + \|W\|_F^2,$$

where $\|\beta\|_2^2 = \int \beta(\bar{\boldsymbol{b}})^2 d(\bar{\boldsymbol{b}})$ and $\|W\|_F^2 = \sum_t \int \boldsymbol{w}_t(\bar{\boldsymbol{b}})^2 d(\bar{\boldsymbol{b}})$. With these in place, we note that Equation (39) can be expressed as Equation (12) with $B$ constrained to be a diagonal operator and $x_{it}$ as the lifted features $\phi(x_{it})$.

*Proof of Theorem 7.1.* The global minimizer of Equation (39) with $d = \infty$ may have infinite support, so the corresponding value may not be achieved by minimizing (17). However, Theorem 6.1 only requires that the we obtain a learner network with regularized loss less than the regularized loss of the teacher network. Since the teacher network has $d$ neurons, this value is attainable by (17). Thus the finite-size network does not need to attain the global minimum of (39) for Claim C.1 to apply.

Since Theorem 6.1 has *no dependence (even in the logarithmic terms)* on the input dimension of the data, it can be applied when the input features the infinite-dimensional feature vector $\phi(\boldsymbol{x})$. The only part of the proof of Theorem 6.1 specific to the nuclear norm is that the dual norm is the operator norm. In Lemma C.5 we had an upper bound on $\frac{1}{n}\|X^\top Z\|_2$. Since we use the $\|\cdot\|_{2,1}$ norm, we must upper bound $\frac{1}{n}\|X^\top Z\|_{2,\infty}$, the dual of the $(2,1)$-norm. Note that $\|A\|_{2,\infty} \le \|A\|_2$, so the upper bound in Lemma C.5 still applies. Thus, Theorem 7.1 follows from Theorem 6.1. $\qquad\square$

*Proof of (21).* The test error of (20) is given by

$$\mathbb{E}[\mathrm{ER}(f_{\hat{B},\hat{\boldsymbol{w}}})] \lesssim \sigma \frac{1}{2\sqrt{n}}(\|B_{T+1}^*\|^2 + \|\boldsymbol{w}_{T+1}^*\|_2^2)\mathbb{E}_{\substack{\boldsymbol{x}_i \overset{\mathrm{n\,iid}}{\sim} p,\\ \boldsymbol{z}\sim N(0,\sigma^2 I)}}[\|\Phi(X)^\top \boldsymbol{z}\|_\infty], \qquad (41)$$

via the basic inequality (c.f. proof of Claim C.2 and C.4). By the matrix Bernstein inequality (c.f. Lemma C.5 or Wei et al. (2019)), $\mathbb{E}_{\boldsymbol{x}_i \overset{\mathrm{n\,iid}}{\sim} p, \boldsymbol{z}\sim N(0,I)}[\|\Phi(X)^\top \boldsymbol{z}\|_\infty] \lesssim \sqrt{\mathrm{tr}(\Sigma)}$. When $B_{T+1}^*, \boldsymbol{w}_{T+1}^*$ are sampled from the same distribution as the source tasks, then $\frac{1}{2}(\|B_{T+1}^*\|^2 + \|\boldsymbol{w}_{T+1}^*\|_2^2) \ge \frac{R}{\sqrt{T}}$. Thus we conclude

$$\mathbb{E}[\mathrm{ER}(f_{\hat{B},\hat{\boldsymbol{w}}})] \lesssim \sigma \frac{R}{\sqrt{T}}\sqrt{\frac{\mathrm{tr}(\Sigma)}{n_2}}.$$

$\qquad\square$

---

[8]Informally if $\alpha \in \mathbb{R}^{D\times T}$ with $D$ potentially infinite, $\|\alpha\|_{2,1} = \min_{\alpha=\mathrm{diag}(b)W} \frac{1}{2}\|b\|_2^2 + \frac{1}{2}\|W\|_F^2$.

