# OpenReview forum: "Few-Shot Learning via Learning the Representation, Provably"
_ICLR.cc/2021/Conference — ICLR 2021 Poster_

### Official Review · AnonReviewer4 · 2020-10-28

**Rating:** 6
**Confidence:** 4

**Review:**

#######################################################################

Summary:
This paper studies the benefit of few-shot learning for sample complexity, when all the tasks (both source and target task) share the same underline representation. Under some assumptions on the data and tasks, this paper improves the previous result based on the iid task assumption and shows that they can utilize all source data. The considered models include linear model (both low dimensional and high dimensional representation) and two-layer neural network.

#######################################################################

Pros:
- Understanding when and why few-shot learning is useful is an important and interesting problem. This work provides some insights into this problem and proves that in some setting few-shot learning is indeed helpful for the sample complexity.
- By introducing some new assumptions on data and tasks which is different from previous the iid assumption on the tasks, the paper shows that one can use all source data.
- Several models are considered in the paper, including linear model and two-layer neural network model.
- The results are mainly focused on the statistical analysis, that is the property of minimizer. The algorithms to achieve such minimizer are also discussed, though for neural net, it may need exponential width.
- The paper is easy to follow and clearly discusses the meaning of assumptions.

##################################################################

Cons/Questions:
- I was wondering if author(s) could discuss [1] in the related work, which seems to focus on the similar problem.
- In Section 4 (low dimensional linear representation), a deterministic target risk is discussed in Remark 4.1. That is, the risk for any fixed target task instead of the average of all target task. I was wondering if such result could also hold in the setting of later sections, i.e., high-dimensional linear representation and two-layer neural network.

[1] On the theory of transfer learning: The importance of task diversity. arXiv preprint arXiv:2006.11650

####################################################################

Minor:
- In Theorem 4.1, \kappa is defined, but does not seem to be used in the theorem statement.

#####################################################################

Thanks for the  response from authors! I will keep my score.

---

> ### Author Response · Authors · 2020-11-17
> **Response to Reviewer 4**
>
> Thank you for your valuable comments and for appreciating our work. Please see below for our answers to your questions.
>
> ------ Discussing [1] ------
>
> The main result in [1] is similar in spirit to our result on general nonlinear representations (Appendix B). Their result is based on explicit control of capacity, and therefore it cannot be applied to the high-dimensional linear representation (Section 5) and *overparametrized* neural network (Section 6) settings considered in our paper. We will add the citation to the paper; thanks for pointing out. We would also like to point out that this paper was published after August 2, 2020 (in NeurIPS 2020), so it should be considered as concurrent work according to the ICLR guidelines (https://iclr.cc/Conferences/2021/ReviewerGuide).
>
> ------ Relax the target task assumption in high-dim sections? ------
>
> Yes, it is possible to generalize the result to allow different target task distributions or deterministic target task. Namely, in Theorem 5.1, if the uncentered covariance matrix of $\theta^\ast_{T+1}$ satisfies $\mathbb{E}[\theta^\ast_{T+1}(\theta^\ast_{T+1})^\top] \preceq \alpha \Theta^*(\Theta^*)^\top/T$, the final bound will incur another factor of $\alpha$. This can be obtained by a slight modification in the proof. We will add this to the paper.
>
>
> ------ $\kappa$ in Theorem 4.1 not used? ------
>
> Notice that $\kappa$ appears in the logarithm in the bound in Theorem 4.1.

---

### Official Review · AnonReviewer2 · 2020-10-28
**Theoretical justification for the usefulness of having more source data in few shot learning**

**Rating:** 7
**Confidence:** 4

**Review:**

## Summary

The paper describes excess risk bounds for ERM in the few-shot learning setting where one has access to $n_1$ samples from each of the $T$ source tasks and $n_2$ samples from the target class. Under the assumption that there exists a ground-truth representation map, shared across all tasks, and a ground-truth task-specific linear map, that maps from the representation space to output space, the authors derive bounds that make use of all $n_1T$ data samples from the source tasks to bound the excess risk of the ERM.

The bounds that are derived are for the following three settings: 1) low dimensional linear representation, 2) general linear map with $\ell$-2 norm-based capacity control for the representation, and 3) linear map with a Relu non-linearity based representation. The key innovation is the structure imparted onto the input distribution of the source and target tasks where the assumption is that the input distribution of *all* the tasks covers the target task. The assumptions get progressively stricter for each of the three cases where for the low dimensional linear case, the assumption is that covariance of the target tsk is dominated by the covariance of all the source tasks, for the high dimensional case, the mean covariance is shared across tasks and finally for the neural network cases where the tasks share the same input distribution. These assumptions allow the improvement of the bound in Maurer et al. 2016.


## Strengths:

1. Overall the paper is well written and motivated. The implication of the results are well discussed and makes for a good read.
2. Under the assumptions on input distribution, results are an improvement over the iid task case and provides theoretical motivation for few-shot learning.
3. Generalizes the Excess risk bound for the ERM in [1] for the low dimensional linear and the high dimensional linear setting. Extends the result to the ERM of 2 layer Relu network.

## Weaknesses

1. The bound is valid when one has access to oracle ERM. This is not the case for most non-convex optimization. Especially in the case of the neural network extension.
2. As pointed out by the authors, many results are very similar to concurrent work of [1]
3. The paper lacks discussion with some prior work  [2] where the authors obtained results for task averaged excess risk. The result in this work is not for new task but since the authors make strong assumptions on input data distribution and task model the result seems like a natural extension of the results in [1].

## Discussion and concerns

Both Theorem 5.1 and 6.1 make the coherence assumption on the target task model. I understand that it allows us to get a bound in terms of $\| \Theta^*\|$ but the implications of this assumption is not well discussed in the paper. This looks like assumption 4.3 for Theorem 4.1. Any understanding of how the results would change if we don’t make the assumption 4.3 and the coherence assumptions?


__EDIT: Added references__
[1] Nilesh Tripuraneni, Chi Jin, and Michael I Jordan. Provable meta-learning of linear representations. arXiv preprint arXiv:2002.11684, 2020
[2] Pontil, M., & Maurer, A. (2013, June). Excess risk bounds for multitask learning with trace norm regularization. In Conference on Learning Theory (pp. 55-76).

---

> ### Author Response · Authors · 2020-11-17
> **Response to Reviewer 2**
>
> Thank you for your valuable comments and for appreciating our work. Please see below for our response to your comments.
>
>
> ------ Access to oracle ERM ------
>
> First, we'd like to point out that our results do not crucially rely on global minimization. If the optimization problem is not optimally solved, the analysis will still go through as is and the gap to the minimum will appear in the final bound. For example, for the low-dim representation learning setting in Section 4, suppose that the optimization problem (7) is solved with an optimality gap:
> $$\frac{1}{2n_1T} \lVert Y - \mathcal{X}(\hat{B}\hat{W}) \rVert_F^2 =  \min_{B, W} \left\\{ \frac{1}{2n_1T} \lVert Y - \mathcal{X}({B}{W}) \rVert_F^2 \right\\} + \mathsf{gap} .$$
> Then the final excess risk bound in Theorem 4.1 will simply incur an additional term $\frac{\mathsf{gap}}{c}$:
> $$\sigma^2 \left( \frac{kd \log(\kappa n_1)}{cn_1T} + \frac{k+\log\frac1\delta}{n_2} \right) + \frac{\mathsf{gap}}{c}.$$
> This follows almost immediately from our proof -- we just need to keep track of $\mathsf{gap}$ in (22). In fact, as long as we can guarantee $$\frac{1}{{n_{1}}T} \lVert \mathcal{X}(\hat{B}\hat{W}) - \mathcal{X}(B^\ast W^\ast) \rVert_F^2 \le \widetilde{\mathsf{gap}}$$ (to replace (22)), we will have the final excess risk bound $$ \frac{\widetilde{\mathsf{gap}}}{c}  + \sigma^2 \frac{k+\log\frac1\delta}{n_2} .$$Similar changes apply to other sections. We will add this note to the final version.
>
> Second, notice that the non-convex optimization problems can indeed be solved globally for most settings considered in our paper. We mentioned in Remark 5.1 and Footnote 6 that the optimization problems for high-dim linear representation and the neural network settings can be solved. For the low-dim linear representation setting in Section 4, the nuclear norm convex relaxation [1] solves it to sufficient accuracy for our bounds to apply.
>
> [1] Negahban and Wainwright. Estimation of (near) low-rank matrices with noise and high-dimensional scaling.
>
>
>
> ------ Similarity to [1] ------
>
> We assume that [1] is the paper by Tripuraneni et al. ("Provable Meta-Learning of Linear Representations"). Yes, as we discussed in the paper, [1] only considered low-dim linear representation learning (with stronger assumptions) and did not consider all the other settings (high-dim linear, neural network, general nonlinear) in our paper. Our paper is considerably more comprehensive.
>
>
> ------ Discussing [2] ------
>
> We are not sure what [2] is. Could you provide the reference? Then we will follow up.
>
>
> ------  The task coherence assumption ------
>
> It is possible to relax the coherence assumptions and pay an "incoherence cost" in the final risk bound. For Theorem 4.1, it is easy to see in the proof that the final bound is proportional to $\lVert \mathbb E_{w\sim\nu}[ww^\top] \rVert$ (from Assumption 4.4) and inversely proportional to $\sigma_k^2(W^*)$ (from Assumption 4.3). Then the bound stated in the theorem can be obtained by plugging in Assumptions 4.3 and 4.4, which represent a benign scenario. Similarly, the coherence assumption in Theorem 5.1 can be changed to assuming the uncentered covariance matrix of $\theta_{T+1}^*$ satisfies $\mathbb{E}[\theta^\ast_{T+1}(\theta^\ast_{T+1})^\top] \preceq \alpha \Theta^\ast(\Theta^\ast)^\top/T$, and then the final bound incurs another factor of $\alpha$. We will add these discussions to the paper; thanks for the question!

---

> > ### Author Response · Authors · 2020-11-21
> > **Discussion of the reference**
> >
> > Thank you very much for pointing out the reference. We notice that there are some fundamental differences between our work and [2].
> >
> > As you noted, [2] only looked at the multitask learning setting and only considered the average excess risk on the training tasks; there is no new task in their setting. In fact, [2] **did not consider representation learning**, and only tried to learn different linear predictors for different tasks. As a result, it was unclear from [2] what can be transferred to a new task, which is the main focus in our paper.
> >
> > That said, we agree that [2] is related to our Section 5 since nuclear norm is used to control the complexity in both settings. We will add the discussion to our paper.

---

### Official Review · AnonReviewer3 · 2020-10-29
**A solid theory paper on few-shot learning**

**Rating:** 8
**Confidence:** 3

**Review:**

The paper aims at justifying the success of few shot learning methods that work based on finding a shared representation among a number of tasks. A serious theoretical challenge is that, even if we assume such a representation exists (and belongs to a predefined class of functions with controlled capacity), we would still need to assume something that connects the source tasks with the target task. Previous work has considered "i.i.d. tasks", however, the obtained bounds were not natural in the sense that we don't have the usual decrease in the error as we increase the size of the training set of the source tasks. Under a different set of assumptions, the authors show that, in a sense, one can "fully" exploit the training data from the source tasks.

Multiple settings are considered, including linear least squares with a low dimensional shared representation, generalization to non-linear representation, high-dimensional (low norm) representation, and neural networks. The obtained results are interesting, and some intuitions are provided about the assumptions and the results. These discussions are sometimes short/dense, making it hard for the reader to follow the details. This is perhaps due to the page-limit.

Perhaps a basic intuition for the first result (linear case with a low dimensional shared representation) is that if the source tasks are somewhat "uniform" in the low dimensional representation and the number of tasks is large enough, then in a sense they will "cover" the low dimensional space, and the learned representation will be in a sense accurate in all directions; so if the target tasks is also selected somewhat uniformly, then the learned representation will work well. Is this correct? In any case, I suggest that the authors add more of these intuitions to the paper, especially for the other problems including the neural net and the high-dimensional linear representation.

It will be interesting to do some experiments on a real world data set and see the extent to which the assumptions are realistic. It will help to see the actual value of the upper bounds on a synthetic data set that adheres the assumptions.

---

> ### Author Response · Authors · 2020-11-17
> **Response to Reviewer 3**
>
> Thank you for your valuable comments and for appreciating our work. The intuition you described is correct; we will add more discussions along these lines to the paper.

---

### Official Review · AnonReviewer1 · 2020-10-29
**Reasonable theoretical results under unrealistic conditions**

**Rating:** 6
**Confidence:** 4

**Review:**

This paper presents some new theoretical insights into a two-layer (linear or non-linear) network based meta-learning framework for dimension reduction and few-shot linear regression. In the considered problem setting, the hidden layer for feature extraction is assumed to be shared across the training and test tasks, and the output layer is optimized in a task-specific way with quadratic loss. For well-specified low-dimensional linear representation learning models, statistical analysis shows that when the tasks are sufficiently divergent, the excess risk of the target task estimator has a near-optimal rate of convergence, up to a near-optimal statistical error of meta-training. The corresponding results for well-specified high-dimensional linear representation and neural networks have also been derived under additional regularization conditions.

Strong points:

-S1. Theoretical understanding of meta-learning for feature extraction and few-shot linear regression is an interesting and timely topic in representation learning.

-S2. The excess risk bounds look correct and can reasonable justify the benefit of multi-task feature extraction in terms of sample efficiency.

-S3. The paper is well organized and neatly presented

Weak points:

-W1. The entire analysis was made under a fundamental assumption that each training task should be globally minimized. From the perspective of optimization, such an assumption is fairly unrealistic in the sense that the linear/non-linear representation learning in the meta-training phase is non-convex and highly non-trivial for optimization. It is thus debatable whether excess risk, which is usually studied in convex learning theory,  should be used as a measurement of generalization performance for the considered representation learning problem.

-W2. In Section 4, it is not very clear how to handle the identifiability issue in model (6), namely, $\hat B$ and $\hat W$ can be properly re-scaled and rotated without changing their product. The same concern can be raised for the underlying true model of data generalization in Equation (4) with linear feature map. The excess risk bounds in Theorem 4.1 and Theorem 5.1 seem to be invariant to such a scaling issue. However, the Assumptions 4.3 and 4.4 are clearly sensitive the scale of model. Is there any way to justify this kind of gap between assumption and result?

-W3. Assumption 4.3 essentially requires that $T\ge k$ because otherwise the smallest singular value will be zero. What will happen if $T<k$? I encourage the authors to provide some discussions/clarifications on this point.

---

> ### Author Response · Authors · 2020-11-17
> **Response to Reviewer 1**
>
> Thank you for your valuable comments and for appreciating our work. Below we address all your concerns. Please let us know if there are any further questions.
>
> ------ W1: assumption of global minimization ------
>
> First, we'd like to point out that our results do not crucially rely on global minimization. If the optimization problem is not optimally solved, the analysis will still go through as is and the gap to the minimum will appear in the final bound. For example, for the low-dim representation learning setting in Section 4, suppose that the optimization problem (7) is solved with an optimality gap:
> $$\frac{1}{2n_1T} \lVert Y - \mathcal{X}(\hat{B}\hat{W}) \rVert_F^2 =  \min_{B, W} \left\\{ \frac{1}{2n_1T} \lVert Y - \mathcal{X}({B}{W}) \rVert_F^2 \right\\} + \mathsf{gap} .$$
> Then the final excess risk bound in Theorem 4.1 will simply incur an additional term $\frac{\mathsf{gap}}{c}$:
> $$\sigma^2 \left( \frac{kd \log(\kappa n_1)}{cn_1T} + \frac{k+\log\frac1\delta}{n_2} \right) + \frac{\mathsf{gap}}{c}.$$
> This follows almost immediately from our proof -- we just need to keep track of $\mathsf{gap}$ in (22). In fact, as long as we can guarantee $$\frac{1}{{n_{1}}T} \lVert \mathcal{X}(\hat{B}\hat{W}) - \mathcal{X}(B^\ast W^\ast) \rVert_F^2 \le \widetilde{\mathsf{gap}}$$ (to replace (22)), we will have the final excess risk bound $$ \frac{\widetilde{\mathsf{gap}}}{c}  + \sigma^2 \frac{k+\log\frac1\delta}{n_2} .$$ Similar changes apply to other sections. We will add this note to the final version.
>
> Second, notice that the non-convex optimization problems can indeed be solved globally for most settings considered in our paper. We mentioned in Remark 5.1 and Footnote 6 that the optimization problems for high-dim linear representation and the neural network settings can be solved. For the low-dim linear representation setting in Section 4, the nuclear norm convex relaxation [1] solves it to sufficient accuracy for our bounds to apply.
>
> [1] Negahban and Wainwright. Estimation of (near) low-rank matrices with noise and high-dimensional scaling.
>
>
> ------ W2: identifiability and scaling ------
>
> It's correct that $\hat B$ and $\hat W$ can be re-scaled/rotated without changing the product. Notice that the product is all we care about, because the final predictor is the linear function described by the product $\hat B \hat w_{T+1}$. The specific scales of $\hat{B}$ and $\hat w_{T+1}$ don't matter as long as their product is unchanged. For example, if the $\hat{B}$ we obtain is 100 times larger, then the corresponding $\hat{w}_{T+1}$ from (8) will be 100 times smaller, and thus the final predictor will still be the same. Therefore Theorem 4.1 and Theorem 5.1 are invariant to the scaling issue.
>
> Also notice that only the ratio between $\lVert \mathbb{E}_{w\sim\nu}[ww^\top] \rVert$ and $\sigma_k^2(W^*)$ (in Assumptions 4.3 and 4.4) impacts the bound in Theorem 4.1. The bound is proportional to $\lVert \mathbb{E}_{w\sim\nu}[ww^\top] \rVert$ and inversely proportional to $\sigma_k^2(W^*)$. This can be seen in the proof on page 16. The scales in Assumptions 4.3 and 4.4 are made to be compatible with the assumption $\lVert w^\ast_t \rVert=\Theta(1)$, but this assumption doesn't affect the final bound and is only made for better illustration. The scale of excess risk is controlled by the noise level $\sigma$ rather than the scale of the ground truth model $B^*$ and $w_t^*$'s. We will add this note to the final version.
>
> We hope this clarifies the confusion.
>
> ------ W3: $T\ge k$ is required? ------
>
> Yes, our result requires $T\ge k$. This representation learning approach would not work well for $T<k$ because it's not possible to identify a $k$-dimensional subspace using fewer than $k$ tasks (at least one of the dimensions would be arbitrarily bad). We will add this clarification; thanks for pointing out.

---

### Decision · Program_Chairs · 2021-01-07
**Final Decision**

**Decision:**

Accept (Poster)

**Comment:**

The paper considers the problem of learning a new task with few examples by using related tasks which can exploit shared representations for which more data is available. The paper proves a number of interesting (primarily theoretical) results.